# TEXT AS ANY-MODALITY FOR ZERO-SHOT CLASSIFICATION BY CONSISTENT PROMPT TUNING

## ABSTRACT

The integration of prompt tuning with multimodal learning has shown significant generalization abilities for various downstream tasks. Despite advancements, existing methods heavily depend on massive modality-specific labeled data (e.g., video, audio, and image), or are customized for a single modality. In this study, we present **T**ext as **A**ny-**M**odality by **C**onsistent **P**rompt **T**uning (TaAM-CPT), a scalable approach for constructing a general representation model toward unlimited modalities using solely text data. TaAM-CPT comprises modality prompt pools, text construction, and modality-aligned text encoders from pre-trained models, which allows for extending new modalities by adding prompt pools and modality-aligned text encoders. To harmonize the learning across different modalities, TaAM-CPT designs intra- and inter-modal learning objectives, which can capture category details within modalities while maintaining semantic consistency across different modalities. Benefiting from its scalable architecture and pre-trained models, TaAM-CPT can be seamlessly extended to accommodate unlimited modalities. Remarkably, without any modality-specific labeled data, TaAM-CPT achieves leading results on diverse datasets spanning various modalities, including video classification (Kinetic-400/600/700), image classification (MSCOCO, VOC2007, NUSWIDE, VOC2012, Objects365), and audio classification (ESC50, US8K). The code is available at `https://anonymous.4open.science/r/TaAM-CPT-0EA6`.

## 1 INTRODUCTION

As unified architectures (Vaswani et al., 2017; Dosovitskiy et al., 2021; Arnab et al., 2021; Gong et al., 2021) and multimodal pre-training models (Devlin et al., 2019; Radford et al., 2021; Tong et al., 2022) progress, recent works have exhibited impressive representation abilities in multimodal learning (Li et al., 2023b; Zhang et al., 2023a; Zhu et al., 2024; Yeh et al., 2023; Wu et al., 2023b). In scenarios restricted by either labeled data or computational resources, owing to the aligned pre-trained models (Radford et al., 2021; Wang et al., 2024b; Wu et al., 2023b), prompt tuning (Zhu et al., 2023a; Yao et al., 2023; Wu et al., 2023a) showcases robust generalization capabilities across various downstream tasks by adjusting a negligible number of parameters, such as video classificatiot (Li et al., 2023a; Wasim et al., 2023), image classification (Zhou et al., 2022b; Hu et al., 2023; Guo et al., 2023), and audio classification (Duan et al., 2024; Chang et al., 2023).

Despite prompt tuning emerging as a novel paradigm for adjusting large-scale pre-trained models (Radford et al., 2021; Dosovitskiy et al., 2021; Wu et al., 2023b), current techniques still rely heavily on massive modality-specific labeled data (e.g. video, audio, and image). For instance, as illustrated in Figure 1 (a) and Figure 1 (e), image supervised methods (Zhou et al., 2022b;a; Sun et al., 2022; Hu et al., 2023) design text prompt that is combined with the textual labels to align with labeled image data for image classification tasks. Likewise, for video and audio classification tasks, previous methods (Ju et al., 2022; Li et al., 2023a; Liu et al., 2024; Kushwaha & Fuentes, 2023) primarily focus on adapting pre-trained multimodal models to video and audio understanding tasks supervised with labeled video and audio data. However, sufficient modality-specific labeled data necessitates considerable manual effort, which, in the face of labeled data limits, can impede the development of robust object classification networks. In the absence of labeled data altogether, these techniques may even fail outright.

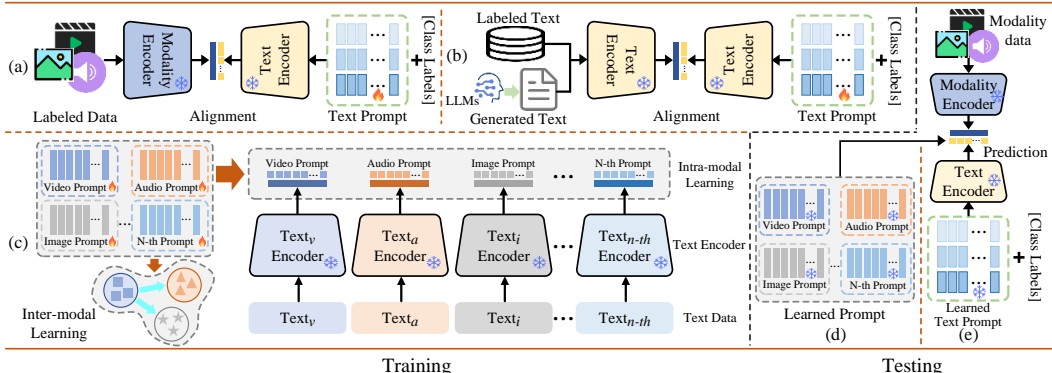

Figure 1: Different prompt tuning paradigms by frozen pre-trained encoders. (a)(b). Supervised methods with labeled and text data. (c). TaAM-CPT. Prompt tuning toward unlimited modalities without prompt encoding processes. (d). Testing of TaAM-CPT. (e). Testing of previous works.

To address the above issue, some studies advocate using the well-aligned embedding space, achieved by contrastive learning (e.g.,CLIP (Cherti et al., 2023)) for prompt tuning. For example, TAI-DPT (Guo et al., 2023), as a pioneering work depicted in Figure 1 (b) and Figure 1 (e), proposes to enable labeled text data (e.g., coco-caption (Lin et al., 2014)) instead of labeled image data for training text prompt, while testing with images and learned text prompt. Similarly, PT-Text (Li et al., 2024) pioneers the approach of audio-free prompt tuning by pre-trained audio-language model (Wu et al., 2023b), where the prompt is learned from text rather than audio for zero-shot audio classification. To further reduce the cost of obtaining labeled text data, PVP (Wu et al., 2024) and TAI-Adapter (Zhu et al., 2023b) recommend using synthetic text data, generated by large language models (LLMs) (Touvron et al., 2023a), as a substitute for labeled text data. However, these strategies require the design of sophisticated text prompt, visual prompt, or adapter frameworks, as well as the deployment of a text encoder to encode the prompts. Additionally, these approaches focus solely on a single modality (e.g., video classification, image classification, or audio classification), and for more modalities, multiple independent models need to be trained additionally.

In this paper, we explore a universal representation model capable of scaling to unlimited modalities without any modality-specific labeled data. This necessitates the following conditions: 1) The model exclusively relies on easy-collected text data for training, eliminating the need for any labeled data. 2) The model architecture needs to be flexible enough to accommodate new categories or modalities and simplify the design of prompt, thereby reducing the complexity of prompt encoding. 3) The model must ensure learning across different modalities does not mutually affect each other, and appropriate training objectives should be designed to enhance the representational capabilities of all modalities.

Motivated by these factors, as shown in Figure 1 (c) and Figure 1 (d), we propose **T**ext as **A**ny-**M**odality for **C**onsistent **P**rompt **T**uning (TaAM-CPT), a general representation model toward unlimited modalities solely using text data generated by LLMs. Unlike TAI-DPT (Guo et al., 2023) and PVP (Wu et al., 2024), which require intricate, multi-grained text prompt designs, our method simplifies the design by characterizing any modality category as a *randomly initialized* vector. Leveraging the instruction following ability of LLMs (Touvron et al., 2023a), we can comfortably obtain text training data for any category. By directly optimizing the vectors within the aligned space of pre-trained models (Radford et al., 2021; Wu et al., 2023b; Wang et al., 2024b), we eliminate intermediate encoding processes. Since the initialization way for each category is identical, TaAM-CPT ensures the flexible addition of any category from any modality without retraining the already learned class-specific prompt. Moreover, we design uni-directional contrastive loss, which uses modalities with stronger representational abilities to guide the learning of those weaker ones. Surprisingly, not only does it enhance the representational abilities of weaker modalities but also further improves the representational abilities of stronger modalities.

We conduct extensive experiments across multiple modalities, including video, audio, and image classification tasks. Without any labeled data, TaAM-CPT achieves superior performance to pre-trained models and recent SOTAs (Guo et al., 2023; Li et al., 2024; Wu et al., 2024). Notably, in image classification, TaAM-CPT outperforms CLIP (Cherti et al., 2023) by 12.5% on MSCOCO (Lin et al., 2014), 8.0~12.0% on Object365 (Shao et al., 2019) and NUSWIDE (Chua et al., 2009). For video

recognition, the top-1 accuracy on K400/K600/K700 (Carreira & Zisserman, 2017; Carreira et al., 2018; 2019) is 1.0~3.0% higher than ViCLIP (Wang et al., 2024b). In audio classification, TaAM-CPT also outperforms the pre-trained model CLAP (Wu et al., 2023b) on ESC50 (Piczak, 2015) and US8K (Salamon et al., 2014). Moreover, our model can be easily integrated with other models that require labeled data for training, thereby further enhancing their classification performance.

## 2 RELATED WORK

**Video, Image, and Audio Classification.** Video classification involves identifying actions in the video. Early works (Wang et al., 2016b; Tran et al., 2018; Feichtenhofer, 2020) focus on designing two-stream networks and 3D CNNs for action recognition. Building on the success of transformers in the image, recent works (Yan et al., 2022b; Xue et al., 2022; Yu et al., 2022; Wang et al., 2022; Li et al., 2023d) explores effective objectives for adapting pre-trained image models to video understanding. To handle the problem of local video redundancy, UniFormerV2 (Li et al., 2022) introduces local and global relation aggregators to learn discriminative representations.

Image classification aims to recognize all the categories in an image. To explore the correlations among labels, some works propose to incorporate semantic dependencies via object proposals (Wang et al., 2016a; Liu et al., 2018), semantic graph (Zhang et al., 2023b; Zhu et al., 2023c), and transformer-based architecture (Bhatti et al., 2023; Scheibenreif et al., 2023). When labeled data is limited, another line of works (Liu et al., 2022b; Simon et al., 2022; Liu et al., 2023b) attempts to solve more challenging scenarios, including zero-shot, few-shot, and partial-label tasks. DualCoOp (Sun et al., 2022) and DualCoOp++ (Hu et al., 2023) learn multiple prompts for each class, resulting in improved performance for both zero-shot and partial-label image classification.

Audio classification involves tagging audio signals into different categories. Traditional works (Henaff et al., 2011; Nanni et al., 2017) mainly rely on machine learning technology and manual feature extraction. In recent years, driven by advancements in deep learning, some works (Xu et al., 2023; Sarkar & Etemad, 2023) have begun to explore the application of neural networks. Additionally, some efforts (Liu et al., 2023a; Garg et al., 2024) attempt to apply the transformer to the audio classification, thereby capturing the long-term dependencies.

**Prompt Tuning in Multimodal Learning.** Prompt tuning (Zhou et al., 2022b; Li et al., 2023a; Duan et al., 2024; Wang et al., 2024a) has emerged for rapidly adapting to downstream tasks by adjusting a minimal number of parameters. For instance, some works (Zhou et al., 2022b; Nie et al., 2023) introduce learnable context vectors to align with images via frozen CLIP encoders. When labeled data is limited, TAI-DPT (Guo et al., 2023) and PT-Text (Li et al., 2024) introduced multi-grained text prompts, surpassing pre-trained multimodal models in image and audio classification tasks solely training text data. PVP (Wu et al., 2024) further enhances image classification performance by co-learning pseudo-visual prompt and text prompt.

Different from the above prompt learning methods, which require a massive of labeled data, complex prompt design, and are limited to single modality design. Our work eliminates the prompt encoder, scales to unlimited modalities, initializes any categories of any modality with an identical vector, and only uses text data generated by LLMs for prompt learning.

## 3 METHODS

The overview architecture of our proposed TaAM-CPT is illustrated in Figure 2. As shown, TaAM-CPT is designed as a general representation model toward unlimited modalities using only text data for prompt learning, which mainly consists of three parts: a) LLMs-assisted data construction, b) Prompt initializing and modality text encoding, and c) Intra- and inter-modal learning.

### 3.1 LLMS-ASSISTED DATA CONSTRUCTION

We present the process of producing appropriate text training data for given modality class labels. Unlike noun filters used in TAI-DPT (Guo et al., 2023) and PVP (Wu et al., 2024), we construct prompt templates to instruct LLMs to generate text sentences that contain the given labels, as shown in Figure 2. For any given labels, we design the following query template:

**TEMPLATE**: *Making several English sentences to describe a* **{ Modality }**. *Requirements: Generate 5 English sentences! Each sentence should be less than 25 words and includes:* **{ Labels }**,

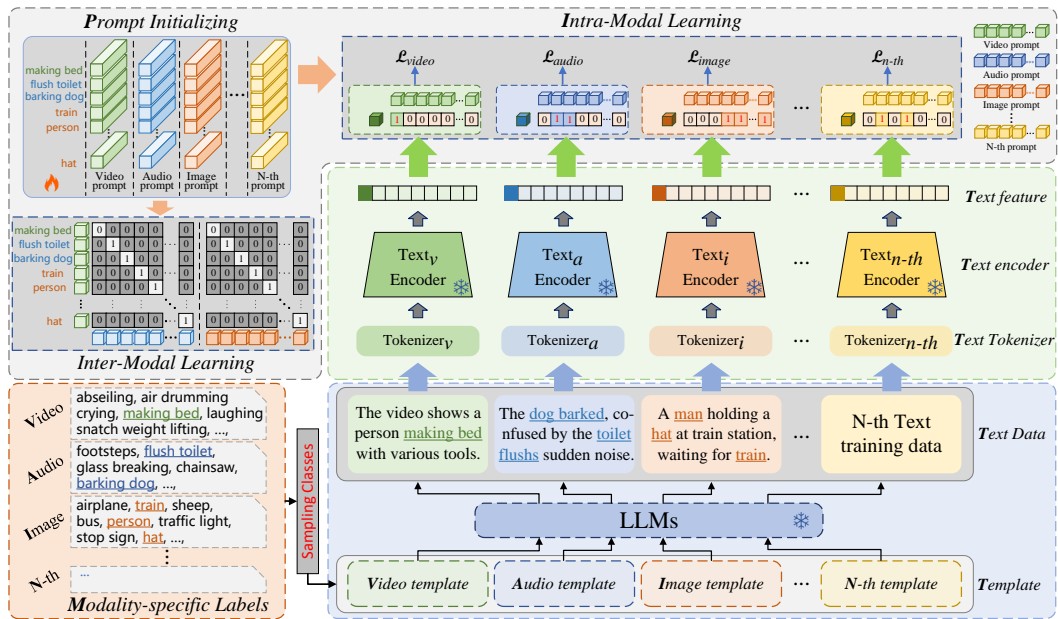

Figure 2: TaAM-CPT overview. We represent any category as a class-specific prompt and use LLMs to generate text data. Intra-modal learning aims to learn each prompt pool by pre-trained models. Inter-modal learning utilizes stronger modalities to guide those weaker ones.

where **{ Modality }** is populated with "video", "audio", "image", etc, and **{ Labels }** indicates modality-specific labels, with a maximum of 2 for video modality, 3 for image and audio modalities. In this way, there are two advantages: the first is to avoid the diversity (e.g., singular and plural) caused by noun filtering, and the second is to avoid the noun filtering to process the phrases describing for video and audio. Therefore, by generating text sentences containing these labels through LLMs, the corresponding ground truth for each sentence is from the **{ Labels }** in the template. More details of prompt templates and text data generated by LLMs are provided in the appendix.

## 3.2 PROMPT INITIALIZING AND MODALITY TEXT ENCODING

**Prompt Initializing.** We take video($\mathcal{V}$), audio($\mathcal{A}$), and image($\mathcal{I}$) modalities as examples to introduce our TaAM-CPT and demonstrate its potential for extension toward unlimited modalities. For each modality, we maintain a modality-specific prompt pool, defined as follows:

$$\mathbf{P}_m = [\ \mathbf{p}_1^m, \mathbf{p}_2^m, \mathbf{p}_3^m, ..., \mathbf{p}_N^m\ ], \tag{1}$$

where $m \in \{\mathcal{V}, \mathcal{A}, \mathcal{I}\}$ represents different modalities; $\boldsymbol{p}_i^m \in \mathbb{R}^d$ denotes $i$-th class-specific prompt; $N$ denotes the total number of labels. Note that the length of the prompt pool is identical for each modality (i.e., $\mathbf{P}_m \in \mathbb{R}^{N \times d}, m \in \{\mathcal{V}, \mathcal{A}, \mathcal{I}\}$), encompassing all labels across different modalities. When a new modality emerges, a new modality-specific prompt pool will be created, avoiding affect the already learned other prompt pools. When a new label arises, a new class-specific prompt will be also added to each prompt pool, avoiding affecting the existing class-specific prompts either. Therefore, TaAM-CPT can be easily extended to unlimited modalities and categories.

**Modality Text Encoding.** According to previous methods (Guo et al., 2023; Li et al., 2024; Wu et al., 2024; Zhu et al., 2023b; Yang et al., 2024), text is treated as a surrogate for other modalities(e.g. image and audio) for zero-shot classification. Such a paradigm potentially assumes that pre-trained models have aligned text with other modalities into a shared embedding space, thereby making it feasible to extract text features as substitutes for other modalities. However, these methods are designed for individual modalities and fail to utilize complementary information among multiple modalities. Hence, as shown in Figure 2, we adopt a parallel architecture and obtain modality-aligned text encoders ($\text{Text}_v, \text{Text}_a, \text{Text}_i$ Encoder) from pre-trained models ViCLIP (Wang et al., 2024b), CLAP (Wu et al., 2023b), and CLIP (Cherti et al., 2023), to extract text features. Furthermore, we find CLIP (Cherti et al., 2023) and CLAP (Wu et al., 2023b) have superior representation abilities for image and audio, compared to ViCLIP (Wang et al., 2024b) for video, specifically reflected in the zero-shot classification performance. Inspired by the discovery, we design an uni-directional

learning strategy to use stronger modalities to guide the learning of weaker modalities. We find that uni-directional learning can improve the performance for all modalities simultaneously.

### 3.3 Intra- and Inter-modal Learning

To learn the modality prompt pool for each modality, our work is to design two learning objectives: a) intra-modal learning aims to optimize the prompt pool for each modality using global text features extracted by modality-aligned text encoders. b) inter-modal learning aims to improve the representational abilities of weaker modalities based on stronger ones.

**Intra-modal Learning.** To make it easier, we take image modality as an example to introduce intra-modal learning, and the same approach is applied to video and audio modalities. The candidate label set is represented as $\mathcal{C} = \{l_1, l_2, ..., l_N\}$, where $N$ is the total number of labels across all modalities. Then, we denote the text training data for image labels as $\mathcal{T} = \{t_i, \mathbf{y}_i\}_{i=1}^{M}$, where $M$ is the number of texts; $\mathbf{y}_i = \{y_{i1}, y_{i2}, ..., y_{i,N}\}$ denotes the ground truth of the text $t_i$ and $y_{ij}$ for $j \in \{1, 2, ..., N\}$ is 1 if the $t_i$ is generated from the label $l_j$ and 0 otherwise. Then, the text embedding of $t_i$ is extracted by frozen text encoder of CLIP (Cherti et al., 2023), formulated as follows:

$$\mathbf{h}_i = \phi(t_i), \tag{2}$$

where $\phi$ denotes the text encoder of CLIP, $\mathbf{h}_i \in \mathbb{R}^d$ denotes the normalized global text feature of $t_i$ with $d$ being the dimension. When processing the input text data of video or audio modalities, we simply replace $\phi$ as the text encoder of ViCLIP (Wang et al., 2024b) or CLAP (Wu et al., 2023b) to extract the corresponding text feature. The similarity of $t_i$ and the prompt pool of image modality can then be computed by:

$$s_{ij} = \langle \mathbf{h}_i, \ \mathbf{p}_j \rangle, \quad \forall j \in \{1, 2, 3, ..., N\}, \tag{3}$$

where $\mathbf{p}_j$ denotes the $j$-th prompt in the prompt pool of image modality. Note that the prompt can be optimized directly without processing through any encoder or MLP. Compared to (Guo et al., 2023; Yang et al., 2024; Li et al., 2024; Wu et al., 2024), which requires the design of complex multi-grained prompt and cumbersome encoding procedure, our method simplifies the design of the prompt and reduces the computational cost to half. For the optimization of prompt, we employ Ranking loss instead of InfoNCE or Cross-Entropy loss, since InfoNCE loss requires massive negative samples and high-cost softmax function to optimize well, Cross-Entropy loss only optimizes positive labels while ignoring loss from negative labels, leading to very slow convergence. Therefore, we employ Ranking loss to directly compare the similarity between positive and negative labels:

$$\mathcal{L}_{\mathbf{I}} = \frac{1}{B} \sum_{k=1}^{B} \sum_{i \in \{c^+\}} \sum_{j \in \{c^-\}} \max(0, m - s_{ki} + s_{kj}), \tag{4}$$

where $c^+$ denotes positive labels with $y_{ij}$ for $j \in \{1, 2, ..., N\}$ is 1, $c^-$ denotes negative labels, $s_{ki}$ and $s_{kj}$ are positive pair and negative pair similarities described in Eq. (3), $m$ is denoted as the margin to measure the difference between each pair of similarities. For the video and audio modalities, we substitute the text encoder $\phi$ described in Eq. (3) to the text encoder of ViCLIP and CLAP to obtain the text feature, and then compute the similarities between the text feature and video prompt pool, audio prompt pool. As a result, we can obtain the Ranking loss $\mathcal{L}_{\mathbf{V}}$ and $\mathcal{L}_{\mathbf{A}}$ and $\mathcal{L}_{\mathbf{I}}$. The total loss for intra-modal learning can be written as:

$$\mathcal{L}_{\mathbf{intra}} = \mathcal{L}_{\mathbf{I}} + \mathcal{L}_{\mathbf{V}} + \mathcal{L}_{\mathbf{A}}. \tag{5}$$

During training, we fix text encoders and optimize the modality-specific prompt pools by Eq. (5). Note that the positive labels in Eq. (4) only contain positive image labels, while negative labels contain not only negative image labels but also labels from other modalities. Other modality's labels serving as negative labels not only expand the number of negative pairs but also enhance the representational ability of video modality. By analogy, this rule can be applied to audio and image modalities also.

**Inter-modal Learning.** Contrastive learning aims to align different modalities, such as image-text, video-text, and audio-text, into a shared embedding space. However, the discrepancy in the information content of image, audio, and video modalities results in a significant modality gap between the aligned video and text modalities and subpar zero-shot classification performance. Motivated by this phenomenon, we propose uni-directional contrastive learning, which guides the learning of weaker modalities using the stronger ones. In this paper, we adaptively determine the

weak modality during training based on the lowest validation performance. Specifically, the video modality is treated as weak as its performance is always lower, and image and audio as stronger ones. To facilitate understanding, we rephrase Eq. (1) into the follow format:

$$\mathbf{P}_{\mathcal{V}} = [\ \mathbf{p}_{v_1}^{\mathcal{V}}, \mathbf{p}_{v_2}^{\mathcal{V}}, ..., \mathbf{p}_{v_v}^{\mathcal{V}}, \mathbf{p}_{a_1}^{\mathcal{V}}, \mathbf{p}_{a_2}^{\mathcal{V}} ..., \mathbf{p}_{a_a}^{\mathcal{V}}, \mathbf{p}_{w_1}^{\mathcal{V}}, \mathbf{p}_{w_2}^{\mathcal{V}}, ..., \mathbf{p}_{w_w}^{\mathcal{V}}\ ],$$
$$\mathbf{P}_{\mathcal{A}} = [\ \mathbf{p}_{v_1}^{\mathcal{A}}, \mathbf{p}_{v_2}^{\mathcal{A}}, ..., \mathbf{p}_{v_v}^{\mathcal{A}}, \mathbf{p}_{a_1}^{\mathcal{A}}, \mathbf{p}_{a_2}^{\mathcal{A}} ..., \mathbf{p}_{a_a}^{\mathcal{A}}, \mathbf{p}_{w_1}^{\mathcal{A}}, \mathbf{p}_{w_2}^{\mathcal{A}}, ..., \mathbf{p}_{w_w}^{\mathcal{A}}\ ], \tag{6}$$
$$\mathbf{P}_{\mathcal{I}} = [\ \mathbf{p}_{v_1}^{\mathcal{I}}, \mathbf{p}_{v_2}^{\mathcal{I}}, ..., \mathbf{p}_{v_v}^{\mathcal{I}}, \mathbf{p}_{a_1}^{\mathcal{I}}, \mathbf{p}_{a_2}^{\mathcal{I}} ..., \mathbf{p}_{a_a}^{\mathcal{I}}, \mathbf{p}_{w_1}^{\mathcal{I}}, \mathbf{p}_{w_2}^{\mathcal{I}}, ..., \mathbf{p}_{w_w}^{\mathcal{I}}\ ],$$

where $v+a+w=N$, $\mathbf{p}_k^{\mathcal{V}}$, $\mathbf{p}_k^{\mathcal{A}}$ and $\mathbf{p}_k^{\mathcal{I}}$ represent class-specific prompt of video, audio, and image prompt pools. Note that the initialized prompt pool of each modality is identical, which means the prompt pool of the video modality contains video labels of size $v$, audio labels of size $a$, and image labels of size $w$. The prompt pool for image and audio modalities is the same as the video modality.

We then present the uni-directional contrastive objective based on $\mathbf{P}_{\mathcal{V}}$ and $\mathbf{P}_{\mathcal{A}}$. Specifically, the similarity matrix can be computed by $\mathbf{P}_{\mathcal{A}}^{\top}\mathbf{P}_{\mathcal{V}} \in \mathbb{R}^{N \times N}$. And the ground truth for $\mathbf{P}_{\mathcal{V}}$ and $\mathbf{P}_{\mathcal{A}}$ of $N$ labels is a diagonal matrix. Note that the size of the similarity matrix and ground truth matrix is batch-size agnostic and equals the number of total labels. Therefore, for each video prompt of $\mathbf{P}_{\mathcal{V}}$ and audio prompt of $\mathbf{P}_{\mathcal{A}}$, the softmax-normalized video prompt to audio prompt and ground truth matrix can be defined as:

$$p_{ij}^{v2a} = \frac{\exp\left(s(\boldsymbol{v}_i, \boldsymbol{a}_j)/\tau\right)}{\sum_{k=1}^{N} \exp\left(s(\boldsymbol{v}_i, \boldsymbol{a}_k)/\tau\right)}, \quad \mathbf{y}^{v2a} = \begin{pmatrix} 0 & 0 & \cdots & 0 & \cdots & 0 \\ 0 & \ddots & \ddots & \vdots & & \vdots \\ \vdots & \ddots & 0 & 0 & \cdots & 0 \\ 0 & \cdots & 0 & 1 & \cdots & 0 \\ \vdots & & \vdots & \vdots & \ddots & \vdots \\ 0 & \cdots & 0 & 0 & \cdots & 1 \end{pmatrix}, \tag{7}$$

where $\boldsymbol{v}_i$ and $\boldsymbol{a}_j$ denote video prompt and audio prompt, $s(\cdot, \cdot)$ represents similarity function, $\tau$ is a learnable temperature parameter. Note that the ground truth label $\mathbf{y}^{v2a}$ is different from the label matrix in vanilla contrastive learning (*i.e.* identity matrix), where the first $v + w$ diagonal elements are set to 0. It indicates that the loss generated at these positions will be ignored when calculating the cross-entropy loss. Therefore, the uni-directional contrastive loss for $\mathbf{P}_{\mathcal{V}}$ and $\mathbf{P}_{\mathcal{A}}$ can be defined as $\mathcal{L}_{v2a} = \mathcal{L}_{CE}(y^{v2a}, p^{v2a})$, where $y_{ij}^{v2a} \in \{0, 1\}$ for $\forall i, j \in \{1, 2, \ldots, N\}$ represents the similarity ground truth between video prompt $\mathbf{v}_i$ and audio prompt $\mathbf{a}_j$. Similarly, we can obtain the uni-directional contrastive loss between the prompt pool of video modality $\mathbf{P}_{\mathcal{V}}$ and prompt pool of image modality $\mathbf{P}_{\mathcal{I}}$: $\mathcal{L}_{v2w} = \mathcal{L}_{CE}(y^{v2w}, p^{v2w})$. And the total inter-learning loss can be defined as:

$$\mathcal{L}_{\mathbf{inter}} = \mathcal{L}_{v2a} + \mathcal{L}_{v2w}. \tag{8}$$

Consequently, We align the prompts of image labels in the video prompt pool with those in the image prompt pool, and the prompts of audio labels with those in the audio prompt pool. These aligned image and audio prompts will be treated as negative labels for training video prompt pool in intra-modal learning, thereby expanding the number of negative pairs. In addition, the diagonal elements corresponding to video and image labels in the ground truth matrix are set to 0, which avoids affecting the learning of the prompt of video labels. During training, we apply uni-directional contrastive learning to video-to-audio and video-to-image. The total loss of TaAM-CPT is: $\mathcal{L}_{\mathbf{total}} = \lambda_1 \mathcal{L}_{\mathbf{intra}} + \lambda_2 \mathcal{L}_{\mathbf{inter}}$, where $\lambda_1$ and $\lambda_2$ denote the loss weights of intra-modal learning and inter-modal learning.

### 3.4 DISCUSSION

In this subsection, as illustrated in Figure 3, we discuss how intra- and inter-modal learning works well. For inter-modal learning, we employ uni-directional contrastive learning, aligning "negative image/audio labels" from the "video prompt pool" with "positive/negative image labels" from the "image prompt pool" and "positive/negative audio labels" from the

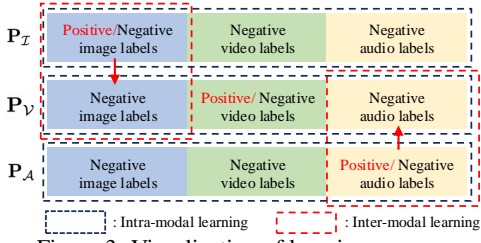

Figure 3: Visualization of learning process.

"audio prompt pool". We can actually treat this process as transferring knowledge from "image/audio prompt pool" to "video prompt pool". For intra-modal learning, take the image modality as an example. Although the "negative labels" contain "negative image/video/audio labels" in the "image prompt pool", these "negative video/audio labels" don't require modality alignment. The purpose

is just to increase the number of negative samples, thereby learning more robust representations of "positive image labels". For the video modality, the "negative labels" come from aligned "negative image/audio labels" and "negative video labels" in the "video prompt pool". Such a uni-directional contrastive learning strategy ensures that "negative image/audio labels" in the "video prompt pool" can not affect the learning of "positive image/audio labels" in the "image/audio prompt pool".

### 3.5 Model Testing

After learning the prompt pool of each modality, each prompt uniquely represents a specific class. We take the video modality as an example to showcase the video classification. Given an input video, we replace the video modality-specific text encoder in Figure 2 with the video encoder of ViCLIP to obtain the video feature. Then, we directly calculate the similarity between the video feature and each prompt in the video prompt pool, and the prediction of the input video is the prompt with the highest similarity. It can be seen that each prompt is calculated directly with the video features without any encoding processing, which significantly improves the inference speed of the model. For image classification and audio classification, we adopt the same approach, calculating the similarity between image or audio and their corresponding prompt pools to obtain the predictions.

## 4 Experiments

### 4.1 Experimental Setup

**Datasets.** We conduct extensive experiments on 13 datasets across video, image, and audio modalities. For video classification, we select UCF101 (Soomro et al., 2012) and large-scale datasets Kinetic-400/600/700 (Carreira & Zisserman, 2017; Carreira et al., 2018; 2019). For image classification, besides MSCOCO (Lin et al., 2014), VOC2007 (Everingham et al., 2010) and NUSWIDE (Chua et al., 2009) used in previous works (Guo et al., 2023; Wu et al., 2024), we also select the VOC2012 (Everingham et al., 2010), ImageNet-mini (Russakovsky et al., 2015) and Objects365 (Shao et al., 2019) to evaluate our method. For audio classification, we follow PT-Text (Li et al., 2024), selecting ESC50 (Piczak, 2015) and US8K (Salamon et al., 2014). For all the datasets mentioned above, we use the official test set to evaluate our method, when the labels of test set are not publicly available, we choose the validation set for evaluation instead. *See appendix for the details of these datasets.*

**Implementation Details.** We select the pre-trained models, open-sourced by the LAION (Schuhmann et al., 2022), as the modality-specific encoders, i.e., ViCLIP-Base (Wang et al., 2024b) for video modality, CLIP-ViT-B-32 (Cherti et al., 2023) for image modality, and CLAP (Wu et al., 2023b) for audio modality. The LLaMA-2-7B (Touvron et al., 2023b) is selected for generating 100k text sentences for each modality, on a single Tesla V100, it takes about 2 hours. By simply adding some spatial relationships instruction in the template, LLaMA-2-7B can generate text descriptions that accurately reflect spatial relationships. For each class-specific prompt, we initialize it as a vector with a length of 512, mean being 0, and std being 0.02. During training, all modality-aligned text encoders are fixed, and only prompts are optimized. We evaluate our methods by top-1/5 accuracy and mean average precision (mAP) metrics. *See appendix for the more implementation details.*

### 4.2 Results on Zero-Shot Tasks

To evaluate TaAM-CPT, besides the zero-shot performance comparison with pre-trained multimodal models (*i.e.* ViCLIP (Wang et al., 2024b), CLIP (Cherti et al., 2023), CLAP (Wu et al., 2023b)), we also compare its performance with existing SOTA methods on image classification and audio classification tasks. Notably, in the zero-shot video classification field, there has been no research that explores a similar training setting, *i.e.*, solely training with text data for prompt tuning. Therefore, we only select ViCLIP (Wang et al., 2024b) as the zero-shot benchmark for comparison.

**Video Classification.** We adopt the default prompt "a video of a [CLASS]" to obtain the zero-shot results of ViCLIP (Russakovsky et al., 2015). From Table 1, our approach outperforms ZS-ViCLIP by 2.1% top-1 and 2.4% top-5 accuracy on UCF101. On the larger Kinetic-400/600/700 datasets with 400, 600,

Table 1: Results with ZS-ViCLIP on zero-shot video classification.

| Methods | UCF101 | | K400 | | K600 | | K700 | |
|---|---|---|---|---|---|---|---|---|
| | top1 | top5 | top1 | top5 | top1 | top5 | top1 | top5 |
| ZS-ViCLIP[ICLR24] | 73.3 | 93.3 | 53.8 | 78.7 | 52.0 | 78.4 | 43.5 | 68.6 |
| **TaAM-CPT**(Ours) | **75.4** | **95.7** | **55.2** | **80.4** | **52.9** | **80.1** | **46.0** | **71.1** |

and 700 labels, respectively, TaAM-CPT also surpasses ZS-ViCLIP by 0.9~3.0% top-1 and top-5 accuracy on all datasets, showing the effectiveness of TaAM-CPT without labeled video data.

Table 2: Comparison with ZS-CLIP and SOTAs on zero-shot image classification.

| Methods | MSCOCO | VOC2007 | VOC2012 | NUSWIDE | ImageNet-mini | Objects365 |
|---|---|---|---|---|---|---|
| ZS-CLIP[ICLR24] | 55.6 | 80.5 | 80.1 | 37.1 | ( 85.5, 94.3 ) | 19.8 |
| TAI-DPT[CVPR23] | 65.1 | 88.3 | 85.1 | 46.5 | ( 86.2, 94.7 ) | 24.1 |
| TAI-Adapter[arXiv23] | 67.7 | 89.0 | 85.5 | **53.3** | ( 86.7, 94.4 ) | 25.8 |
| Data-free[arXiv24] | 66.8 | 88.7 | 86.0 | 47.0 | ( 86.1, 94.9 ) | 23.9 |
| PVP[IJCAI24] | 67.7 | 88.9 | 86.2 | 49.3 | ( 87.4, 95.3 ) | 26.3 |
| **TaAM-CPT**(Ours) | **68.1** | **89.4** | **87.8** | 49.6 | ( **90.4**, **98.3** ) | **28.2** |

**Image Classification.** For zero-shot image classification, we present the results in Table 2 and compare our approach with SOTAs TAI-DPT (Guo et al., 2023),TAI-Adapter (Zhu et al., 2023b), Data-free (Yang et al., 2024), and PVP (Wu et al., 2024) trained with complex prompt design or adapter module. The results of ZS-CLIP are obtained by inputting the default prompt "a photo of a [CLASS]" to CLIP. From Table 2, our TaAM-CPT outperforms ZS-CLIP by a large margin of 12.5% and 12.4% mAP on MSCOCO and NUSWIDE, respectively. On VOC2007 and VOC2012 with 20 object classes, our method also improves by 7.0% ~9.0% over ZS-CLIP. For large-scale datasets, TaAM-CPT can still achieve promising results over ZS-CLIP, e.g., 90.4% *vs* 85.5% top-1 accuracy on ImageNet-mini, and 28.2% *vs* 19.8% mAP on Objects365. Compared with these SOTAs that also solely train with text data, our method achieves sota performance in most datasets, while the previous methods require the design of complex prompt and prompt encoding processes.

**Audio Classification.** The results for zero-shot audio classification with CLAP (Wu et al., 2023b) and recent SOTA PT-Text (Li et al., 2024) are shown in Table 3. Our TaAM-CPT outperforms ZS-CLAP with 3.7% and 9.0% accuracy on ESC50 (Piczak, 2015) and US8K (Salamon et al., 2014), despite the high performance of CLAP. Furthermore, without intricate prompt design, TaAM-CPT surpasses PT-Text 0.3% on the ESC50 dataset.

Table 3: Results on zero-shot audio classification.

| Methods | ESC50 | US8K |
|---|---|---|
| ZS-CLAP[ICASSP23] | 90.5 | 76.2 |
| PT-Text[ICASSP24] | 93.9 | – |
| **TaAM-CPT**(Ours) | **94.2** | **85.2** |

## 4.3 INTEGRATING WITH OTHER METHODS

Following TAI-DPT (Guo et al., 2023), we conduct the experiments of integrating TaAM-CPT with other supervised models in an off-the-shelf manner, further improving their performance. Take a video with $n$ labels as an example, the supervised model's softmax predictions denote as $\mathbf{P_S} = (\mathrm{p}_{s1}, \mathrm{p}_{s2}, ..., \mathrm{p}_{sn})$. For TaAM-CPT, we calculate the similarity between video and $n$ class-specific video prompts and obtain softmax predictions $\mathbf{P_T} = (\mathrm{p}_{t1}, \mathrm{p}_{t2}, ..., \mathrm{p}_{tn})$. Therefore, the intergrated results can be computed by $\mathbf{P_I} = (\mathrm{p}_{s1} + \mathrm{p}_{t1}, \mathrm{p}_{s2} + \mathrm{p}_{t2}, ..., \mathrm{p}_{sn} + \mathrm{p}_{tn})$.

**Video Classification.** We select the Base size model of Video Swin Transformer (Liu et al., 2022a), MTV (Yan et al., 2022a), AIM (Yang et al., 2023), UniFormerV2 (Li et al., 2022), and UMT (Li et al., 2023c) as baselines. The results are shown in Table 4. After integrating our TaAM-CPT with Video Swin, MTV, AIM-B, UniFormerV2-B, and UMT-B on Kinetic-400/600/700 datasets, the video classification performance of these methods can be further improved, while these methods achieve promising performances.

Table 4: Results of integrating TaAM-CPT with supervised models on Kinetic-400/600/700 datasets.

| Methods | K400 | | K600 | | K700 | |
|---|---|---|---|---|---|---|
| | top1 | top5 | top1 | top5 | top1 | top5 |
| Video Swin[CVPR22] | 82.7 | 95.5 | 84.0 | 96.5 | – | – |
| +**TaAM-CPT**(Ours) | **83.5** | **95.9** | **84.8** | **97.1** | – | – |
| MTV[CVPR22] | 81.8 | 95.0 | 83.8 | 96.2 | 73.5 | 90.3 |
| +**TaAM-CPT**(Ours) | **82.9** | **95.7** | **84.7** | **97.0** | **74.8** | **91.2** |
| AIM[ICLR23] | 83.9 | 96.3 | – | – | 76.9 | 92.1 |
| +**TaAM-CPT**(Ours) | **84.6** | **97.2** | – | – | **77.2** | **93.0** |
| UniFormerV2[ICCV23] | 84.0 | 96.3 | 84.8 | 96.8 | 75.4 | 92.6 |
| +**TaAM-CPT**(Ours) | **84.8** | **97.1** | **85.5** | **97.6** | **76.1** | **93.4** |
| UMT[ICCV23] | 85.7 | 97.0 | 87.8 | 97.8 | 78.5 | 94.3 |
| +**TaAM-CPT**(Ours) | **86.2** | **97.6** | **88.1** | **98.0** | **78.8** | **94.7** |

**Image Classification.** In Table 5, we select the newest DualCoOp++ (Hu et al., 2023) instead of DualCoOp (Sun et al., 2022) used in previous SOTAs (Guo et al., 2023; Wu et al., 2024), and reproduce DualCoOp++ on these datasets (marked with *). + indicates integrating predictions with DualCoOp++*. In Table 5, while DualCoOp++* obtains promising performance, +TaAM-CPT can further enhance the image classification results. Compared with +TAI-DPT and +PVP, our +TaAM-CPT achieves higher performance in all cases, and surpasses +PVP by considerable margins of 0.2%, 0.3%, and 1.2% mAP on these datasets. Notably, TAI-DPT and PVP rely on costly prompt encoders

Table 5: The mAP results for partial-label setting on all datasets, where the performance of +TAI-DPT/+PVP/+TaAM-CPT integrates the predictions of TAI-DPT/PVP/TaAM-CPT and DualCoOp++*.

| | Methods | 10% | 20% | 30% | 40% | 50% | 60% | 70% | 80% | 90% | Avg |
|---|---|---|---|---|---|---|---|---|---|---|---|
| **MSCOCO** | DualCoOp[NeurIPS22] | 81.0 | 82.3 | 82.9 | 83.4 | 83.5 | 83.9 | 84.0 | 84.1 | 84.3 | 83.3 |
| | DualCoOp++[TPAMI24] | 81.4 | 83.1 | 83.7 | 84.2 | 84.4 | 84.5 | 84.8 | 85.0 | 85.1 | 84.0 |
| | DualCoOp++*[TPAMI24] | 81.5 | 83.2 | 84.0 | 84.4 | 84.5 | 84.7 | 84.8 | 85.1 | 85.2 | 84.1 |
| | +TAI-DPT[CVPR23] | 81.7 | 83.3 | 84.5 | 84.5 | 84.7 | 85.0 | 85.1 | 85.2 | 85.2 | 84.3 |
| | +PVP[IJCAI24] | 82.1 | 83.6 | 84.5 | 84.7 | 85.0 | **85.3** | 85.3 | 85.6 | 85.6 | 84.6 |
| | **+TaAM-CPT(Ours)** | **82.4** | **83.8** | **84.6** | **85.0** | **85.1** | **85.3** | **85.5** | **85.7** | **85.8** | **84.8** |
| **VOC2007** | DualCoOp[NeurIPS22] | 91.4 | 93.8 | 93.8 | 94.3 | 94.6 | 94.7 | 94.8 | 94.9 | 94.9 | 94.1 |
| | DualCoOp++[TPAMI24] | 92.7 | 93.4 | 93.8 | 94.0 | 94.3 | 94.4 | 94.4 | 94.7 | 94.9 | 94.1 |
| | DualCoOp++*[TPAMI24] | 93.0 | 93.9 | 94.2 | 94.4 | 94.6 | 94.8 | 94.9 | 95.1 | 95.0 | 94.4 |
| | +TAI-DPT[CVPR23] | 93.2 | 94.0 | 94.2 | 94.6 | 94.7 | 94.8 | 95.0 | 95.1 | 95.1 | 94.5 |
| | +PVP[IJCAI24] | 93.5 | 94.3 | 94.4 | 94.6 | 95.0 | 95.1 | 95.2 | 95.2 | 95.3 | 94.7 |
| | **+TaAM-CPT(Ours)** | **93.9** | **94.6** | **94.8** | **95.1** | **95.3** | **95.4** | **95.4** | **95.5** | **95.6** | **95.0** |
| **NUSWIDE** | DualCoOp[NeurIPS22] | 54.0 | 56.2 | 56.9 | 57.4 | 57.9 | 57.9 | 57.6 | 58.2 | 58.8 | 57.2 |
| | DualCoOp++*[TPAMI24] | 54.4 | 56.6 | 58.1 | 58.7 | 58.9 | 59.3 | 59.7 | 59.8 | 60.1 | 58.4 |
| | +TAI-DPT[CVPR23] | 56.9 | 58.1 | 58.5 | 58.8 | 58.8 | 59.1 | 59.1 | 59.5 | 60.0 | 58.7 |
| | +PVP[IJCAI24] | 57.3 | 58.6 | 59.3 | 59.4 | 59.6 | 60.0 | 60.1 | 60.1 | 60.3 | 59.4 |
| | **+TaAM-CPT(Ours)** | **58.2** | **59.6** | **60.5** | **60.7** | **60.8** | **61.3** | **61.4** | **61.3** | **61.7** | **60.6** |

and are only customized for a single image modality. Our TaAM-CPT is a general representation model that can accommodate unlimited modalities and class labels.

**Audio Classification.** We also study the audio classification results of integrating with HTS-AT (Chen et al., 2022) and CrissCross (Sarkar & Etemad, 2023). As the same video classification task, we compute the similarities between the audio feature and audio prompt pool as the predictions. From Table 5, the performance of both HST-AT and CrissCross is enhanced on ESC50 (Piczak, 2015) and US8K (Salamon et al., 2014) datasets.

Table 6: Results of integrating TaAM-CPT with supervised audio classification methods.

| Methods | ESC50 | US8K |
|---|---|---|
| HTS-AT[ICASSP22] | 97.0 | 94.7 |
| **+TaAM-CPT(Ours)** | **97.2** | **95.1** |
| CrissCross[AAAI23] | 90.5 | 92.1 |
| **+TaAM-CPT(Ours)** | **94.7** | **92.8** |

### 4.4 FURTHER ANALYSIS

We conduct further analysis to explore TaAM-CPT. More results (e.g., each component, hyperparameter, prompt dimension, more datasets, training data size, etc.) are presented in appendix.

**Quantity of modalities and categories.** We first explore the feasibility of TaAM-CPT for unlimited modalities and categories. For clarity, we adopt the Bert-base model (Devlin et al., 2019) with *110MB* parameters as reference. TaAM-CPT initializes each prompt with a $512\text{-}d$ vector, meaning one class prompt occupies 512 parameters. Therefore, for $N$ modalities, TaAM-CPT can accommodate approximately $\frac{110,000,000}{512n}$ class prompts. For example, for 10 modalities, the prompt pool size can reach 21484, which is sufficient to cover the common categories.

**Inter-moda Learning.** In Table 7, $\langle a, b \rangle \longrightarrow \langle c \rangle$ denotes uni-directional contrastive learning from $a, b$ to $c$, while $\longleftrightarrow$ denotes naive bi-directional learning. Both $\langle I, V \rangle \longrightarrow \langle A \rangle$ and $\langle A, V \rangle \longrightarrow \langle I \rangle$ improve the performance of image and audio modalities while decreasing on video modality. Notably, $\langle I \rangle \longrightarrow \langle V \rangle$ and $\langle A \rangle \longrightarrow \langle V \rangle$ significantly outperform ZS-CLIP and ZS-CLAP by a large margin, demonstrating the effectiveness of inter-modal learning. Additionally, uni-directional learning can achieve higher performance than bi-directional learning on all datasets.

Table 7: Results of different learning manners.

| $\mathcal{L}_{\text{Inter}}$ | K400 | MSCOCO | ESC50 |
|---|---|---|---|
| ZS-ViCLIP,CLIP,CLAP | ( 53.8, 78.7 ) | 55.6 | 90.5 |
| $\langle I, V \rangle \longrightarrow \langle A \rangle$ | ( 52.1, 79.3 ) | 64.8 | 91.7 |
| $\langle A, V \rangle \longrightarrow \langle I \rangle$ | ( 51.9, 79.5 ) | 65.1 | 91.8 |
| $\langle I \rangle \longrightarrow \langle V \rangle$ | ( 53.6, 79.4 ) | 67.1 | 92.4 |
| $\langle A \rangle \longrightarrow \langle V \rangle$ | ( 53.2, 79.2 ) | 65.3 | 93.2 |
| $\langle I, A \rangle \longleftrightarrow \langle V \rangle$ | ( 54.3, 79.8 ) | 67.1 | 92.9 |
| $\langle I, A \rangle \longrightarrow \langle V \rangle$(Ours) | **( 55.2, 80.4 )** | **68.1** | **94.2** |

**Prompt Initialization.** Here, we explore the initializations of the prompt in Table 8. Different from randomly initializing the prompt in the method, we use the output embeddings by CLIP's text encoder to initialize class-specific prompt and remove inter-modal learning. Therefore, each class-specific prompt encompasses class-specific textual prior knowledge, allowing TaAM-CPT to converge quickly with less training data (we collect only 50 text

Table 8: Results of different prompt initialization.

| Prompt initialization | K400 | MSCOCO | ESC50 |
|---|---|---|---|
| ZS-ViCLIP,CLIP,CLAP | ( 53.8, 78.7 ) | 55.6 | 90.5 |
| Initialize by CLIP, w/o $\mathcal{L}_{\text{Inter}}$ | ( 54.5, 79.6 ) | 65.3 | 93.1 |
| **TaAM-CPT(Ours)** | **( 55.2, 80.4 )** | **68.1** | **94.2** |

training data for one class). Although without inter-modal learning, TaAM-CPT achieves higher performance compared to CLIP, ViCLIP, and CLAP.

## 4.5 VISUALIZATION

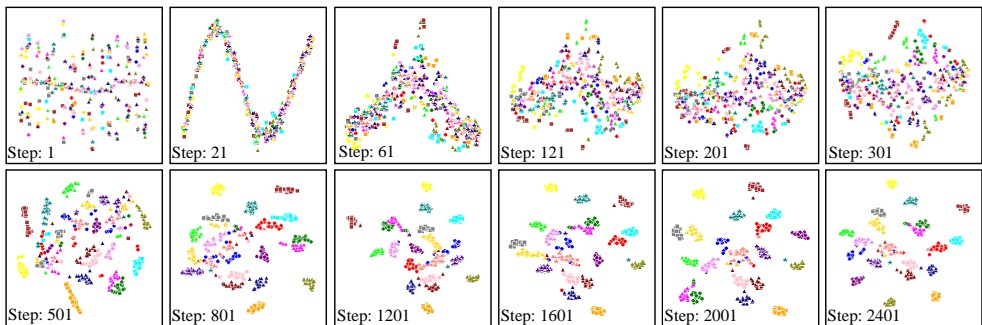

Figure 4: Distribution of video prompt and video feature by t-SNE (van der Maaten & Hinton, 2008).

**Intra-modal Learning.** We randomly selected 20 video classes on Kinetic-400. For each video sample, we computed its similarity with each video prompt, resulting in a 400-d vector and using t-SNE (van der Maaten & Hinton, 2008) for visualization, which reflects the learning process of each video class prompt in Figure 4. Since the initialization method is identical, video samples from the same category show a uniform distribution before model training (*Step: 0*). As training progresses, the class-specific prompt begins to learn the unique representations (*Step: 21~1201*) for each category (*Step: 1601~2401*). *Visualizations for more datasets can be found in the appendix.*

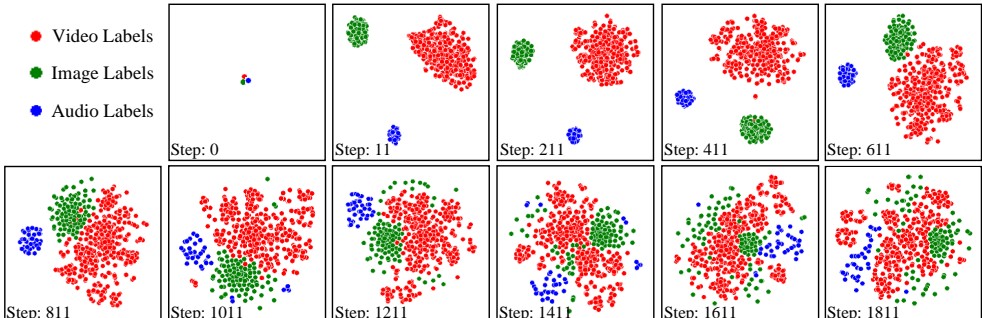

Figure 5: Distribution of prompt for different modalities by t-SNE (van der Maaten & Hinton, 2008).

**Inter-modal Learning.** We select Kinetic-400, MSCOCO, and ESC50 datasets, which contain 400, 80, and 50 class labels, respectively. As shown in Figure 5, before model training (*Step: 0*), the prompt pools for each modality are initialized in the same vector. When starting training, the distribution of different modalities rapidly separates (*Step: 11~211*), as each modality first learns modality-specific representations through modality-aligned text encoders. As training progresses, uni-directional contrastive learning gradually pulls the representation space of the video modality towards image and audio modalities (*Step: 411~1411*), indicating that the video modality is continuously learning the representations of image and audio modalities. Furthermore, each modality still maintains its own representation space without being disrupted by the other modalities (*Step: 1611~1811*).

## 5 CONCLUSION

In this paper, we explore a scalable way of constructing a universal representation model for various modalities. Based on a flexible architecture and aligned pre-trained models, we develop TaAM-CPT, treating any category as a learnable vector and optimizing it directly through aligned pre-trained models. In addition, uni-directional contrastive learning also improves the classification performance of all modalities. The experimental results on 13 datasets show that TaAM-CPT achieves leading results in various classification tasks, including zero-shot video classification, image classification, audio classification, and partial-label image classification.

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

# Appendix for Text as Any-Modality for Zero-Shot Classification by Consistent Prompt Tuning

## A    DETAILS OF PRE-TRAINED MULTIMODAL MODELS.

Our TaAM-CPT is built upon multimodal pre-trained models, including video-language model, image-language model, and audio-language model, and uses frozen text encoders for prompt tuning, as well as frozen modality encoders for object recognition predicting. In our work, we choose the pretrained multimodal models, open-sourced by the LAION (Schuhmann et al., 2022) organization, as the modality-aligned text and modality encoders. For a total of 300k text sentences on a single Tesla V100 for the Kinetic-400, MSCOCO, and ESC50 datasets, each epoch takes 12 minutes and the total training cost for 10 epochs is about 2 hours.

**ViCLIP.** ViCLIP is a video-language pretraining model, building upon the open-source CLIP of OpenAI. The model consists of a video encoder and corresponding text encoder, which is pretrained on the InternVid dataset containing 7 million videos, each with detailed text descriptions. We use the BASE architecture as our baseline model with 12 attention layers and 512 encoding dimensions.

**CLIP.** We select the open-source image-language pretraining model released by the LAION organization as our baseline model. The model comprises an image encoder and corresponding transformer-based text encoder, each with 12 attention layers and an encoding dimension of 512. The size of the input image is $224 \times 224$, with the patch size being 32. For image modality, CLIP-ViT-B-32 (Cherti et al., 2023) is selected as the image encoder and image-text encoder.

**CLAP.** For the audio-language pretraining model, likewise, we select CLAP released by the LAION organization as our baseline model. The audio encoder is a transformer-based model with 4 groups of swin-transformer blocks, while the text encoder is RoBERTa. Two-layer MLPs with ReLU activation are applied to mAP both audio and text outputs into 512 dimensions. For audio modality, we select CLAP (Wu et al., 2023b) from LAION (Schuhmann et al., 2022) as the audio encoder and the built-in Robert as the audio-text encoder.

## B    DETAILS OF DATASETS

### B.1    VIDEO DATASETS

**UCF101.** UCF101 (Soomro et al., 2012) is a commonly used video classification dataset that contains 101 different action classes, each class contains approximately 100~300 video clips, and a total of 13,320 video clips. These video clips are collected from real data on YouTube, ranging in length from 10~30 seconds. We use all of the video data to evaluate our methods.

**Kinetic-400.** Kinetic-400 (Carreira & Zisserman, 2017) is a large-scale, high-quality video dataset collected from YouTube, including 400 human action classes. Each action class contains 450~1150 video clips, covering a wide range of classes, e.g., playing instruments, interactions between humans and objects, and handshakes. Each action has 250~1000 video clips for the training set, 50 video clips for the validation set, and 100 video clips for the test set. The validation set is used to evaluate our methods.

**Kinetic-600.** Kinetic-600 (Carreira et al., 2018) is an extension of the Kinetic-400 dataset, comprising approximately 480K video clips from 600 action classes. Each action class has at least 700 video clips. The dataset consists of 450~1000 video clips for training, 50 for validation, and 100 for testing per action class. The validation set is used to evaluate our methods.

**Kinetic-700.** Kinetic-700 (Carreira et al., 2019) is an extension of the Kinetic-600 dataset, covering 700 human action classes. Each action class has at least 700 video clips. Each video is a 10-second action clip extracted from original YouTube videos and labeled accordingly. There are a total of 650,000 video clips, with each action class comprising 450,100 video clips for training, 5,000 video clips for validation, and 1,000 video clips for testing. We use the validation set to evaluate our methods.

### B.2 IMAGE DATASETS

**MSCOCO.** MSCOCO (Lin et al., 2014) is a large-scale computer vision dataset used for tasks such as object recognition, object detection, and image segmentation. It includes 80 image classes, 328,000 images, and 2,500,000 instances. It comprises 82,783 training images, 40,504 validation images, and 40,775 test images. We use the validation set to evaluate our methods.

**VOC2007.** VOC2007 (Everingham et al., 2010) is an image dataset containing 20 image classes that can be used to evaluate image classification, object detection, and image segmentation tasks. It consists of 9,963 images in total, with 5,011 images in the training set and 4,952 images in the test set. The test set is used to evaluate our methods.

**VOC2012.** VOC2012 (Everingham et al., 2010) dataset contains 20 classes, including people, animals, vehicles, indoor objects, and a background category, making a total of 20 classes. It can be used for evaluating image classification, object detection, and image segmentation tasks. It comprises 11,540 images, with 5,717 images in the training set and 5,823 images in the test set. The test set is used to evaluate our methods.

**NUSWIDE.** NUSWIDE (Chua et al., 2009) is an image dataset that contains 269,648 images collected from Flickr, with a total of 81 manually annotated concepts, including objects and scenes. It includes 161,789 images for the training set and 107,859 images for the validation set. We use the validation set to evaluate our methods.

**ImageNet-mini.** ImageNet-mini (Russakovsky et al., 2015) is derived from the ImageNet dataset and contains 100 classes with a total of 60,000 images, with 600 samples per class. The training and validation sets are typically divided into an 8:2 ratio by class. (For small sample classification, 64 classes are used for training, 16 for validation, and 20 for testing.) We use the test set to evaluate our methods.

**Objects365.** Objects365 (Shao et al., 2019) is a large object detection dataset that contains 638k images, 365 image classes, and 10,101k bounding boxes, far surpassing datasets like COCO. According to the paper's annotation process, a total of 740k images were annotated, with 600k used for training, 38k for validation, and 100k for testing. We use the test set to evaluate our methods.

### B.3 AUDIO DATASETS

**ESC50.** ESC50 (Piczak, 2015) is a standard dataset for environmental sound classification that contains 50 different environmental categories, each with 40 samples of up to 5 seconds in duration, totaling 2,000 samples. These samples cover a wide range of environments, such as animal sounds, traffic noise, indoor activities, etc. All samples are carefully balanced to ensure uniformity when training models. We use the validation set to evaluate our methods.

**US8K.** UrbanSound8k (Salamon et al., 2014) is a widely used open data set for automatic urban environment sound classification, which includes ten categories such as air conditioning sound and car horn sound. There are 8732 audio clips in the dataset with a length of about 4 seconds. The data set is divided into training and testing sets. We use the test set to evaluate our methods.

## C TRAINING TEXT DATA CONSTRUCTION.

Here, we discuss the text training data construction for different modalities. We construct the following prompt template to input into LLaMA-2-7B for generating text description data.

**TEMPLATE**: Make several English sentences to describe a **{ Modality }**. *Requirements: Generate 5 English sentences! Each sentence should be less than 25 words and includes:* **{ Labels }**.

where **{ Modality }** is replaced with video, audio, and image, **{ Labels }** denotes the sampled classes. For video and audio datasets, which typically involve single classification tasks, we set the number of sampled categories to 2 to prevent too many categories from appearing in one sentence, which could interfere with the model's learning of specific representations for each category. For image classification datasets, where multiple categories can appear on a single image, the number of sampled categories is set to 1, 2, 3, or 4 to ensure that the model not only learns the dependencies between image categories but also acquires independent representations for each category. As shown in Figure

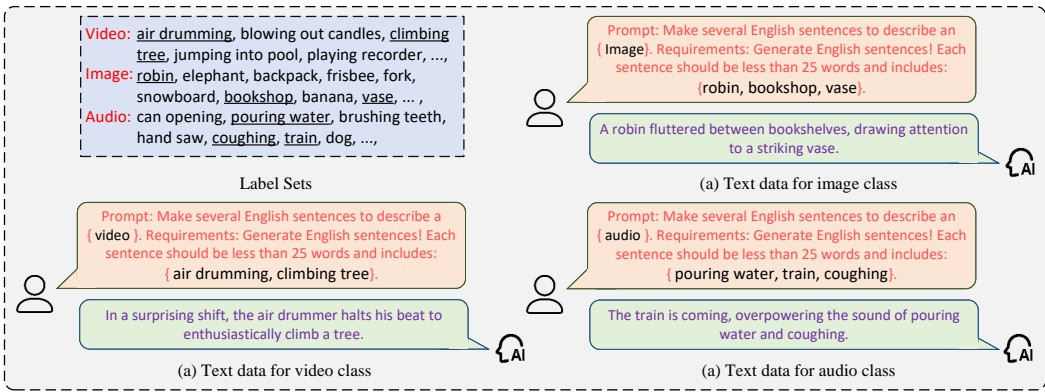

Figure 6: The candidate label set and text data generated by LLMs.

6, we randomly select several classes from the label set and construct a prompt template to query the LLMs to generate text data containing the semantic information of these classes.

# D  ABLATION STUDY

**Prompt Design.** Here, we mainly discuss the variants of consistent prompt tuning (CPT) in Table 9: a) Shared-Intra (1024), where the prompt is initialized as 1024-d vector and mapped to 512-d through a FC; b) Shared-Intra (512) represents initialization as a 512-d vector and then mapped to 512-d; c) Shared-

Table 9: Results of different prompt designs.

| Prompt | K400 | MSCOCO | ESC50 |
|---|---|---|---|
| Shared-Intra (1024) | ( 43.1, 74.2 ) | 55.4 | 90.6 |
| Shared-Intra (512) | ( 47.5, 75.3 ) | 58.7 | 91.9 |
| Shared-Inter (512) | ( 50.1, 79.3 ) | 62.2 | 92.1 |
| **TaAM-CPT**(Ours) | **( 55.2, 80.4 )** | **68.1** | **94.2** |

Inter (512), where all prompts across all modalities share a FC and are mapped to 512-d. On Kinetic-400, we note a pronounced degradation of these variants. We believe the decline is mainly attributable to the numerous categories that are semantically proximate (e.g., *making pizza* and *making sandwich*). These phenomena are also observed in the MSCOCO and ESC50 datasets.

**Unified Architecture.** Our TaAM-CPT is designed as a general model toward unlimited modalities, exhibiting more robust object recognition capabilities than single modality-specific models. Table 7 presents the results of training each modality independently by intra-modal learning (e.g. *VP* ✓*with* $\mathcal{L}_{\mathbf{Ia}}$ ✓), as well as the impact of applying the uni-

Table 10: Results of evaluating the unified architecture.

| VP | IP | AP | $\mathcal{L}_{\mathbf{Ia}}$ | $\mathcal{L}_{\mathbf{Ie}}$ | K400 | MSCOCO | ESC50 |
|---|---|---|---|---|---|---|---|
| ZS-ViCLIP,CLIP,CLAP | | | | | ( 53.8, 78.7 ) | 55.6 | 90.5 |
| ✓ | × | × | ✓ | × | ( 53.8, 78.9 ) | – | – |
| × | ✓ | × | ✓ | × | – | 65.8 | – |
| × | × | ✓ | ✓ | × | – | – | 92.5 |
| ✓ | ✓ | ✓ | ✓ | × | ( 53.7, 79.1 ) | 65.2 | 92.7 |
| ✓ | ✓ | ✓ | ✓ | ✓ | **( 55.2, 80.4 )** | **68.1** | **94.2** |

directional contrastive learning ($\mathcal{L}_{\mathbf{Ie}}$) across modalities. We can see that training single modality prompt by intra-modal learning has already yielded better results than the pre-trained models, and when all modalities are trained together, the performance of each modality can be further improved. In addition, applying uni-directional contrastive learning to guide the learning of video modality, not only improves the performance of the video modality but also enhances the object classification capabilities of the image and audio modalities.

**Loss Weight.** In this study, we design Ranking loss and uni-directional contrastive loss to perform intra-modal learning and inter-modal learning. The Ranking loss aims to learn class-specific prompt for each modality, while the contrastive loss is applied to guide the learning of weaker modalities (video) through those stronger ones (image and audio). Here, we explore the impact of setting different loss weights for these two loss functions. As shown in Figure 11, $\mathcal{L}_{\mathbf{Ia}}$ represents the Ranking loss for intra-modal learning, and

Table 11: Results of different loss weight between intra-modal learning and inter-modal learning.

| $\mathcal{L}_{\mathbf{Ia}}$ | $\mathcal{L}_{\mathbf{Ie}}$ | K400 | MSCOCO | ESC50 |
|---|---|---|---|---|
| 0.4 | 1.6 | ( 54.9, 80.0 ) | 67.9 | 94.0 |
| 0.8 | 1.2 | ( 55.1, 80.2 ) | 68.1 | 94.1 |
| 1.0 | 1.0 | **( 55.2, 80.4 )** | **68.1** | **94.2** |
| 1.2 | 0.8 | ( 55.0, 80.2 ) | 68.0 | 94.0 |
| 1.6 | 0.4 | ( 54.5, 79.6 ) | 68.0 | 93.9 |

$\mathcal{L}_{\mathbf{Ie}}$ represents the uni-directional contrastive loss for inter-modal learning. Our method achieves the best results when the weights of $\mathcal{L}_{\mathbf{Ia}}$ and $\mathcal{L}_{\mathbf{Ie}}$ are identical. Additionally, we notice that when the weight of $\mathcal{L}_{\mathbf{Ie}}$ is set to 1.0,0.8 and 0.4, there is a significant decrease in top-1 and top-5 accuracy on the Kinetic-400 dataset, while the performance on MSCOCO and ESC50 datasets only suffer minor damage. This indicates that inter-modal learning greatly affects the learning of weaker modality, which is the video modality in this case.

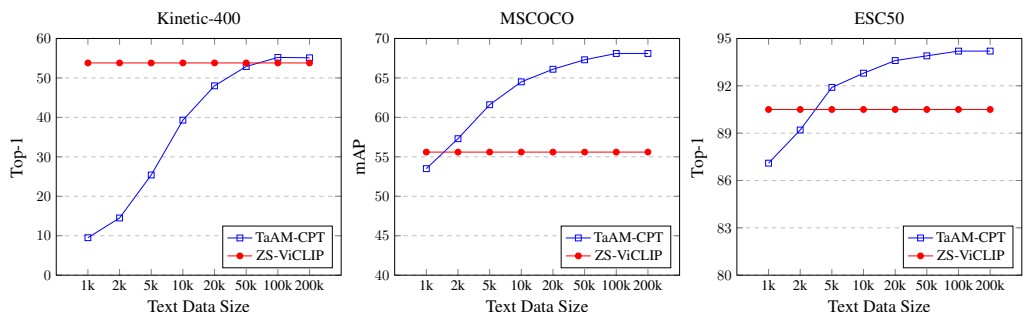

Figure 7: Results of different size of text training data on Kinetic-400, MSCOCO and ESC50 datasets.

**Text Training Data Size.** Our TaAM-CPT is trained with text data generated by LLMs instead of modality-specific labeled data. Therefore, we conduct various experiments with different sizes of text training data on the Kinetic-400, MSCOCO, and ESC50 datasets. As shown in Figure 7, on the Kinetic-400 dataset with text data size being 1k, the top-1 accuracy is only 9.8% due to the insufficient number of text data for each class, which hinders the learning of robust class-specific representations. However, as continuing to expand the scale of text training data, the corresponding text data for each class also increases gradually. When the text data reaches 100K, our TaAM-CPT outperforms ZS-ViCLIP. On the MSCOCO and ESC50 datasets, which contain 80 and 50 class labels, respectively, when the amount of text data is 5K, our method has already significantly surpassed ZS-CLIP and ZS-CLAP by 7% mAP and 2% top-1 accuracy. The performance on these two datasets begins to stabilize when the amount of text data is increased to 50K, indicating that datasets with more classes require a larger scale of text training data.

## E  VISUALIZATION OF INTRA-MODAL LEARNING.

Here, as shown in Figure 8, 9, 10, 11, 12, we present the more visualization results of the distribution of class-specific prompt learned by intra-modal learning on Kinetic-600/700, MSCOCO, ImageNet-mini, and ESC50 datasets.

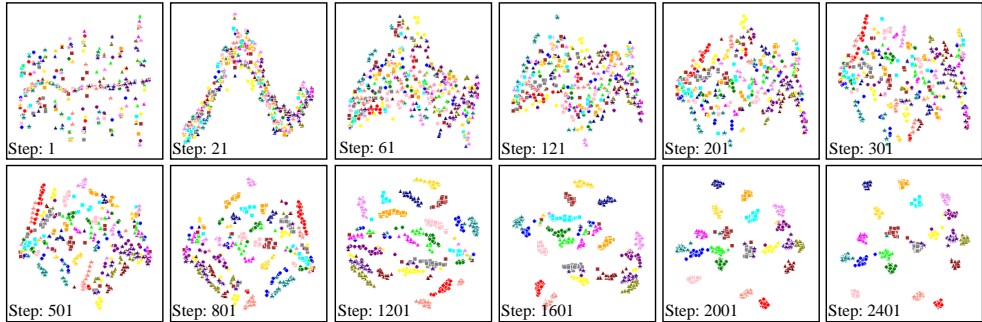

Figure 8: Visualization of the distribution of video prompt and video feature using t-SNE (van der Maaten & Hinton, 2008) for dimensionality reduction. We randomly select 20 video classes from the Kinetic-600 dataset.

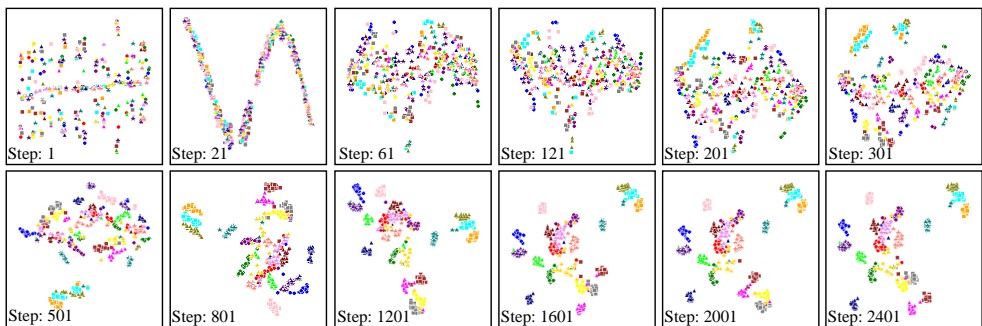

Figure 9: Visualization of the distribution of video prompt and video feature using t-SNE (van der Maaten & Hinton, 2008) for dimensionality reduction. We randomly select 20 video classes from the Kinetic-700 dataset.

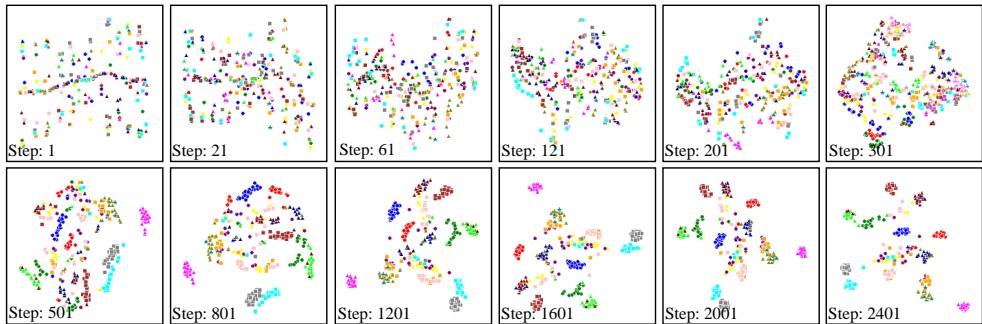

Figure 10: Visualization of the distribution of image prompt and image feature using t-SNE (van der Maaten & Hinton, 2008) for dimensionality reduction. We randomly select 20 image classes from the MSCOCO dataset.

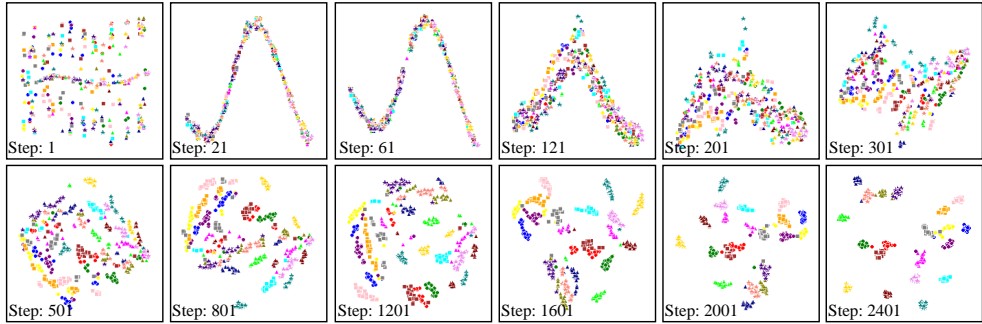

Figure 11: Visualization of the distribution of image prompt and image feature using t-SNE (van der Maaten & Hinton, 2008) for dimensionality reduction. We randomly select 20 image classes from the ImageNet-mini dataset.

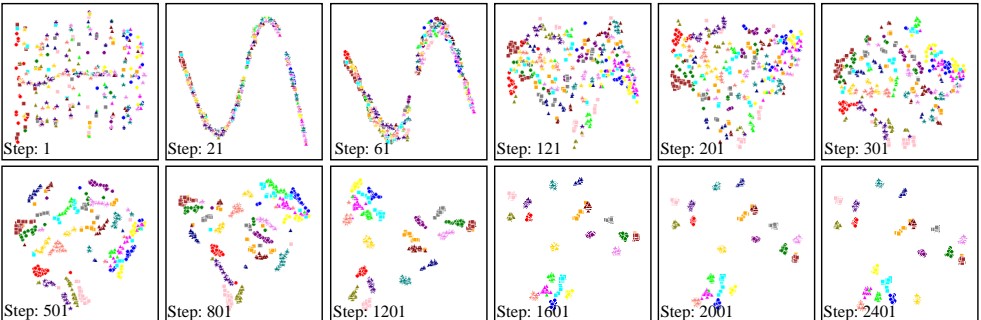

Figure 12: Visualization of the distribution of audio prompt and audio feature using t-SNE (van der Maaten & Hinton, 2008) for dimensionality reduction. We randomly select 20 audio classes from the ESC50 dataset.

