# OpenReview forum: "Text as Any-Modality for Zero-shot Classification by Consistent Prompt Tuning"
_ICLR.cc/2025/Conference — Submitted to ICLR 2025_

### Official Review · Reviewer_5HaF · 2024-10-29

**Soundness:** 3
**Presentation:** 3
**Contribution:** 3
**Rating:** 5
**Confidence:** 4

**Summary:**

This paper introduces TaAM-CPT, a scalable framework for general representation learning across unlimited modalities using only text data. Unlike existing methods that rely on large amounts of modality-specific labeled data or focus on a single modality, TaAM-CPT leverages prompt tuning, modality-aligned text encoders, and intra- and inter-modal objectives to harmonize learning across different modalities. With its flexible architecture, TaAM-CPT achieves top performance in zero-shot and classification tasks across 13 diverse datasets, spanning video, image, and audio classification, without the need for labeled data specific to each modality.

**Strengths:**

1. TaAM-CPT supports multiple modalities without needing labeled data, advancing universal representation learning.
2. The model achieves state-of-the-art results across diverse tasks—zero-shot video, image, and audio classification—demonstrating its robust generalization capabilities across various modalities and datasets, a key advantage for multimodal applications.
3. The organization of this paper is logical.

**Weaknesses:**

1. One of my main concerns is that the proposed TaAM-CPT seems to be a combination of existing techniques, _e.g._, learnable prompts (soft prompts), Inter-/Intra-modal Learning strategies. The authors are expected to provide more discussions to better showcase their novelty and contributions.
2. In Intra-modal Learning, the authors claim that the proposed method _'simplifies the design of the prompt and reduces the computational cost to half'_ (lines 242-243). However, the experimental section fails to adequately demonstrate the efficiency of the proposed approach. It would be beneficial for the authors to include quantitative metrics such as training cost or parameters to substantiate the robustness and flexibility of their approach.
3. The clarity and simplicity of the paper's writing could be enhanced. Certain sentences were perceived as verbose and challenging to comprehend. Notably, lines 51-53 and Eq. 6 would benefit from refinement to improve their accessibility and ease of understanding.

**Questions:**

Please refer to the weaknesses, I will raise my score if all concerns are well-addressed.

---

> ### Author Response · Authors · 2024-11-21
> **Official Response by Authors (1/1)**
>
> Thanks for your valuable suggestions and we will try to address your concerns as follows. We are eager to engage in a more detailed discussion with you.
>
> ### **Weakness 1: More discussions about the novelty and contributions.**
> + Thank you for your valuable feedback during the review process. We have noted your concerns regarding the potential similarities between our TaAM-CPT method and existing techniques. Here is a further elaboration on the novelty of TaAM-CPT:
>     + The core innovation of our work, TaAM-CPT, lies in its minimalist training process, which relies solely on text data without any modality-specific annotated data, further enhancing the applicability of the multimodal pre-trained model in data-scarce scenarios.
>     + Compared to methods such as TAI-DPT and PVP, TaAM-CPT simplifies the prompt design by representing each category as a randomly initialized vector. This design not only reduces the complexity of the model but also allows for more flexibility in adding new categories and modalities without disrupting the learned category prompts.
>     + TaAM-CPT adopts a unidirectional contrastive learning strategy, using modalities with stronger representational capabilities to guide those with weaker learning abilities. This approach not only improves the performance of weaker modalities but also further enhances the representational power of stronger modalities.
>     + TaAM-CPT is entirely dependent on easily collectible text data and does not require annotated data for specific modalities. This gives the method an advantage in situations where annotated data is scarce or prohibitively expensive to obtain.
>
> ### **Weakness 1: Demonstrate robustness and flexibility, such as training cost or parameters.**
> + We appreciate your concerns regarding the applicability. We believe TaAM-CPT is still a recommended choice when pursuing performance on specific task. For clarity, we adopt Bert-base model with 110MB parameters as reference. TaAM-CPT initializes each prompt with a 512-d vector, meaning one class prompt occupies 512 parameters. Therefore, for n modalities, TaAM-CPT can accommodate approximately 110,000,000512n class prompts. For example, for 10 modalities, the prompt pool size can reach 21,484, which is sufficient to cover the common categories.
>
> + Additionally, we conduct new experiments on K400/MSCOCO/ESC50 datasets, as shown in the below table. Different from randomly initializing the prompt in the paper, we use the output embeddings by ~CLIPs text encoder to initialize class prompt and remove inter-modal learning. Therefore, each class prompt encompasses class-specific textual prior knowledge, allowing TaAM-CPT to converge fastly with less training data (50 texts for one class). Although without inter-modal learning, TaAM-CPT achieves higher performance compared to CLIP/ViCLIP/CLAP.
>
> | Training epoch | Training Size | K400 | MSCOCO | ESC50 |
> | -------  | ------- | ------- | ------- | ------- |
> | ZS-ViCLIP,CLIP,CLAP |  | (53.8, 78.7) | 55.6 | 90.5 |
> | Initialize with CLIP, w/o LInter | 50 for one class | (54.5, 79.6) | 65.3 | 93.1 |
> | **TaAM-CPT** |  | **(55.2, 80.4)** | **68.1** | **94.2** |
>
> ### **Weakness 1: Certain sentences were perceived as verbose and challenging to comprehend.**
> + Thank you for your valuable feedback on the clarity and conciseness of the TaAM-CPT paper. We will reorganize lines 51-53 and rephrase Equation (6). Additionally, we will add more explanatory content to the paper, such as definitions of key concepts, detailed descriptions of algorithmic steps, and in-depth analysis of experimental results, to help readers better understand the TaAM-CPT method.

---

> > ### Comment · Reviewer_5HaF · 2024-11-27
> >
> > The authors address some of my concerns. Nevertheless, the rebuttal lacks comprehensive reporting on training or inference costs, which hampers the ability to substantiate the proposed method's claimed efficiency. While the authors articulate the various parameters related to the design of prompts, it remains challenging to assess the efficacy in mitigating computational complexity when compared with previous methods. Therefore, I will keep my score unchanged.

---

> ### Comment · Area_Chair_Q33N · 2024-11-27
>
> Dear reviewer,
>
> Today is the last day for reviewers to ask questions to authors. Did the authors' rebuttal address your concern? Do you have any additional questions?

---

### Official Review · Reviewer_7Qs2 · 2024-11-02

**Soundness:** 1
**Presentation:** 2
**Contribution:** 2
**Rating:** 5
**Confidence:** 3

**Summary:**

This paper presents TaAM-CPT, a prompt tuning approach for classifying a sample from any modality to a set of predefined categories.
It builds on pre-trained text-modality aligning models such as ViCLIP (text-video), CLIP (text-image) or CLAP (text-audio).
Keeping these models frozen, TaAM-CPT tunes a set of prompt pools (one prompt pool per modality) to align to text representations directly in the representation space of the pre-trained models. The prompts are trained using a combination of inter-modal uni-directional contrastive loss and intra-modal ranking loss.
The paper reports performance on video classification, image classification and audio classification, but claims to be extendable to any number of other modalities.

**Strengths:**

Simple approach:
Compared to TaI-DPT (Guo et al., 2023) and follow-up work, the method presented in this paper is simpler as it does not involve complex multi-grained prompts.
TaAM-CPT only uses a single prompt per category per modality which simplifies the approach.

Code availability:
The authors of TaAM-CPT released the code for their implementation which is very valuable for reproducibility.

**Weaknesses:**

Significance of the quantitative improvement:
The inter-modal unidirectional contrastive learning is the main contribution claimed by this paper. It is ablated in Table 7 and Table 10. But without a statistical analysis of the results it is hard to evaluate the significance of the improvement.
L959: "when all modalities are trained together, the performance of each modality can be further improved." This does not seem to be the case though.
Comparing to independently training each modality, training all modalities jointly does not seem significantly better (53.8=>53.7, 65.8=>65.2, 92.5=>92.7)

Poor presentation and clarity:
The problem tackled in this paper is not properly introduced. It is not clearly explained what prompt-tuning is and why it is interesting.
There are many important parts of the method that are not properly explained and remain unclear. See "Questions" section for details.

**Questions:**

Extension to more modalities and labels:
L201: "When a new modality emerges, a new modality-specific prompt pool will be created, avoiding affecting the already learned other prompt pools. When a new label arises, a new class-specific prompt will be also added to each prompt pool, avoiding affecting the existing class-specific prompts either."
The paper claims that the approach is easily extendable to new modalities and new labels, but does not actually experiment with it.
Could the authors further detail what would be the process for adding another modality or a new label? Would it be possible to keep the old prompts frozen and only train the new prompts? Would the performance on the new prompts be as good as if all prompts had been trained from scratch?
How will a new modality encoder with a different output dimension (e.g. 768) be introduced?

Choice of Video as the "Weak" modality:
L213: "Furthermore, we find CLIP (Cherti et al., 2023) and CLAP (Wu et al., 2023b) have superior representation abilities for image and audio, compared to ViCLIP (Wang et al., 2024b) for video, specifically reflected in the zero-shot classification performance."
On which criteria do the authors claim the video modality as "weak"? A poor zero-shot performance does not necessarily mean that the modality is "weak" or difficult to process, it can also mean that the benchmark itself is hard.

Number of labels per query:
In the main paper, the stated number of labels per query is 2 for video and 3 for image and audio:
L187: "{Labels} indicates modality-specific labels, with a maximum of 2 for video modality, 3 for image and audio modalities."
How were these specific values determined? Why not 3 for video and 4 for image? What would be the impact of only using a single label per query? From Figure 6 it seems like some label combinations lead to captions that do not make much sense.
Moreover, there seem to be a discrepancy in the stated number of labels per query: In the appendix, the stated number of labels per query is 2 for video and audio and 1 to 4 for image:
L913: "For video and audio datasets, we set the number of sampled categories to 2. For image, the number of sampled categories is set to 1, 2, 3, or 4"
Can the authors clarify?

Modality-specific labels:
What if there are common labels across modalities, can the authors confirm that they are not merged together? For example, "crying" could be a video, audio or image label. In that case is it present 3 times in the N labels?

Number of categories:
What is the value of the number of categories (v+a+w=N) used in this paper? How many for video, image and audio? Is it a concatenation of all the labels from benchmark datasets?

Unclear ablation study for Prompt Design:
The section "D Ablation Study - Prompt Design" is unclear. Can the authors clarify what is being discussed exactly?
It seems to be about the initial prompts dimension and their projection into a lower-dimensional space but the motivation is not explained.
What does FC stand for? Fully-connected layer? Please remind the reader about the setting adopted by TaAM-CPT in that regard, is it 512-d without transformation?

Minor presentation notes:
In figure 1 and 2, N is used for representing the number of modalities.
But in the text, N represents the number of categories:
L226: N is the total number of labels across all modalities.

Minor typos:
L040: "classificatiot"
L145: "a massive of labeled data"
L410: "intergrated"
L464: "110MB parameters"

---

> ### Author Response · Authors · 2024-11-21
> **Official Response by Authors (1/2)**
>
> Thanks for your valuable suggestions and we will try to address your concerns as follows. We are eager to engage in a more detailed discussion with you.
>
> ### **Question 1: How to add a new modality or label? Is it possible to train only new prompts while freezing the old ones? Will the performance on new prompts be as good as training all prompts from scratch? How to introduce a new modality encoder with a different output dimension (e.g., 768)?**
> + TaAM-CPT learns a class-specific prompt for each label, when a new label appears, only requires initializing a class-specific prompt for it. The pre-trained models selected in our experiments all share uniform embedding dimensions, which avoid other trainable parameters besides the prompt as much as possible. The more trainable parameters there are, the more likely it distorts the original knowledge in the pre-trained model. Therefore, when a new modality encoder has a 768-dimensional space, we can add a simple linear layer to map the dimension of this modality to the unified dimension, and also consider this linear layer as trainable parameters. We will continue to explore related experiments in future work.
> + For simplicity, we use **K400 with 30 video labels** and **MSCOCO with 30 image labels** as the initial label sets. Novel concepts are other labels **(K400 with 20 video labels, MSCOCO with 20 image labels)**, and the novel audio modality **(ESC50 with 20 audio labels)**, and each label generates 500 text data by LLMs. The experimental results are shown below:
>
> | Modality_Labels   |  |  |  |
> | ------- | ------- | ------- | ------- |
> | K400_Normal_30    | Initial training **(58.1, 82.3)** | Continue training **(58.4, 82.5)** | Frozen **(58.1, 82.3)** |
> | MSCOCO_Normal_30  | Initial training **(65.4)** | Continue training **(65.6)** | Frozen **(65.4)** |
> | K400_Novel_20     | Initial training **(56.3)** | Novel training **(56.7)** | Novel training **(56.5)** |
> | MSCOCO_Novel_20   | Initial training **(69.2)** | Novel training **(68.8)** | Novel training **(69.3)** |
> | ESC50_Novel_20    | Initial training **(92.1)** | Novel training **(92.5)** | Novel training **(93.0)** |
> | **Training time** | **28min** | **30min** | **17min** |
>
> + Among them, **(K400_Novel_20, MSCOCO_Novel_20, ESC50_Novel_20)** represent the added new labels and new modality. We can see that for the normal labels **(K400_Normal_30, MSCOCO_Normal_30)**, whether they continue training **(Continue training)** or are frozen **(Frozen)**, neither affects the learning of the new modality labels and maintain the performance that matches the initial joint training **(Initial training)**.
>
> ### **Question 2: Which criteria claim the video modality as "weak"?**
> + During the training of TaAM-CPT, we directly employ validation performance as a simple strategy. In fact, besides validation performance, metrics such as training performance, training loss, convergence speed, etc., can all serve as criteria for determining weak modalities. We assume that these criteria can assess the hard degree of learning a modality, considering modalities that are difficult to learn as weak modalities and expect strong modalities can further guide the learning of weak modalities.
>
> | Training epoch  | K400 | MSCOCO | ESC50 |
> | -------  | ------- | ------- | ------- |
> | Epoch-00 | 487.3 | 135.4 | 53.2 |
> | Epoch-03 | 386.1 | 105.3 | 14.2 |
> | Epoch-06 | 298.7 | 87.5 | 8.7 |
> | Epoch-09 | 287.5 | 83.5 | 6.9 |
>
> + We can see that the training loss for the video modality is consistently the highest, which is due to the strong correlation between some video categories. Therefore, in the benchmarks, the video modality is always considered a weak modality. Moreover, as shown in Table 7 in the manuscript, if image or audio are regarded as weak modalities, it will adversely affect the performance of all modalities.

---

> ### Author Response · Authors · 2024-11-21
> **Official Response by Authors (2/2)**
>
> ### **Question 3: Number of labels per query: How are these specific values determined? Why not use 3 for video and 4 for image? What effect does using only one label per query have? Clarify the number of labels per query.**
> + We apologize for any confusion caused. In our experiments, the setting is as follows: the video modality has a maximum of 2 labels, while the image and audio modalities have a maximum of 3 labels. This is the result we determined through experimentation:
>
> | Numbers of per query  | K400 | MSCOCO | ESC50 |
> | -------  | ------- | ------- | ------- |
> | (video:2, image:3, audio:3) | (55.2, 80.4) | 68.1 | 94.2 |
> | (video:1, image:2, audio:2) | (43.8, 71.9) | 64.2 | 89.7 |
> | (video:1, image:1, audio:1) | (42.6, 69.5) | 59.6 | 87.3 |
>
> + We believe that for the video modality, each video sample typically represents a complex action, such as playing a guitar or riding a motorcycle. When the number of {labels} is too large, it can result in generating longer text data, making it difficult to accurately express the semantics of each action category. For the image and audio modalities, the categories usually represent a specific object. Moreover, an image or audio often contains several different object categories, so a larger number of {labels} can reflect the interdependencies between categories.
>
> ### **Question 4: If there are common labels across modalities, can it confirm that they are not merged?**
> + If common labels appear, each modality needs to have a customized, class-specific prompt. For example, for the "crying" label, TaAM-CPT defines separate prompts, aligning them with the video modality, image modality, and audio modality, respectively. This way, in downstream tasks, it will be possible to perform video classification, image classification, and audio classification tasks separately.
>
> ### **Question 5: What is the value of N (v+a+w=N) in the paper? How many categories for video, image, and audio? Is it a combination of all benchmark dataset labels?**
> + (v+a+w=N) represents the concatenation of all labels from all modality datasets, as shown in Figure 3 of Section 3.4 in the manuscript. Each modality includes all categories from all modalities.
>
> ### **Question 6: Unclear ablation study for Prompt Design. What does FC stand for? Is it 512-d without transformation?**
> + The ablation studies we conducted primarily investigated the impact of different prompt design methods on the performance of TaAM-CPT, including:
>     + Shared internal dimension: We initialized the prompts as 1024-dimensional or 512-dimensional vectors and projected them into a 512-dimensional space through a fully connected layer (FC).
>     + Shared internal dimension (512): We initialized the prompts as 512-dimensional vectors and used 512 dimensions directly as the final dimension.
>     + Shared internal dimension (512) + Shared external dimension: We shared a fully connected layer for all modality prompts and projected them into a 512-dimensional space.
> + We found that when the prompt dimension is high and projection is necessary, the model's performance significantly decreases. This may be due to an excessive number of parameters, which makes it difficult for the model to learn effective feature representations.
> + FC stands for the fully connected layer, which is used to map vectors from a 1024-dimensional space to a unified 512-dimensional space. In TaAM-CPT, we use 512 dimensions as the final dimension for the prompt pool and do not require projection.
>
> ### **Question 7: Minor presentation notes and Minor typos.**
> + Thank you to the reviewer for pointing this out. We will use $N$ consistently throughout to represent the number of categories and use $M$ to indicate the number of modalities. We will correct all spelling and grammatical errors in the future version to ensure that the format adheres to the standards.

---

> ### Comment · Area_Chair_Q33N · 2024-11-27
>
> Dear reviewer,
>
> Today is the last day for reviewers to ask questions to authors. Did the authors' rebuttal address your concern? Do you have any additional questions?

---

### Official Review · Reviewer_25FQ · 2024-11-03

**Soundness:** 3
**Presentation:** 2
**Contribution:** 3
**Rating:** 5
**Confidence:** 4

**Summary:**

The authors explore a generic representation model capable of scaling to an infinite number of modalities without the need for any modality-specific labelled data.TaAM-CPT simplifies the design by characterising any modal category as a randomly initialised vector, and exploits the command tracking capabilities of the LL.M. to allow easy access to textual training data of any category. TaAM-CPT ensures the flexibility to add any category from any modality without retraining the already learned category-specific cues. Additionally, the authors designed a one-way contrast loss, which uses modalities with stronger representational capabilities to guide the learning of those that are weaker.

**Strengths:**

- The authors present a simplistic approach to prompt building and describe the build process in detail.
- The authors propose a   uni-directional contrastive loss to facilitate intermodal training.
- TaAM-CPT effectively integrates the image/audio/video modalities and achieves competitive performance on classification tasks.

**Weaknesses:**

- Making prompts requires inserting **{Label}**, does this mean that different pools of prompts need to be designed for different datasets?
- LLM's hallucinations may affect the quality of prompt generation, does TaAM-CPT have a process for quality checking during prompt production?
- TaAM-CPT needs to adjust Inter-modal Learning based on validation performance, but I noticed that some of the dataset's validation set is being used for evaluation, is there an information leak?
- Based on the previous question, should TaAM-CPT be trained individually based on the validation performance of each dataset?
- Section 3.5 mentions that TaAM-CPT improves the inference speed of the model, whether the authors have experimented on objective metrics?
- The authors utilize LLM to generate prompts for auxiliary training. The method can be taken as a distillation from LLM. Whether the authors have compared the performance difference between TaAM-CPT and MLLM?
- In section 4.4, the author discusses the feasibility of TaAM-CPT for infinite modes and categories, but I still have concerns about this. When creating prompts, it is possible to combine 2-3 labels in various ways. In extreme cases, exhaustively permuting a vast number of labels can be very time-consuming and may negatively impact the quality of the model.
- With more modalities and labels, I concern that the burden of training intra- and inter-modalities will increase.

**Questions:**

See details in Weaknesses.

---

> ### Author Response · Authors · 2024-11-21
> **Official Response by Authors (1/2)**
>
> Thanks for your valuable suggestions and we will try to address your concerns as follows. We are eager to engage in a more detailed discussion with you.
> ### **Weakness 1: Whether different prompts need to be designed for different datasets.**
> + During the text data generation, the construction of the input prompt template for LLMs is only dependent on the **modality** and **categories**. For example:
> + Prompt template for the video labels **{playing the piano, driving the car}**: *"Making several English sentences to describe a $\textbf {video}$. Requirements: Generate 5 English sentences! Each sentence should be less than 25 words and includes: $\textbf {playing the piano, driving the car}$*".
> + Prompt template for the image labels **{dog, cat, person}**: *"Making several English sentences to describe a $\textbf {image}$. Requirements: Generate 5 English sentences! Each sentence should be less than 25 words and includes: $\textbf {dog, cat, person}$*".
>
> ### **Weakness 2: Can TaAM-CPT check prompt quality during prompt generation?**
> + For the text data generation strategy in PVP [1], after obtaining a text description, PVP [1] would again query LLMs whether the description is reasonable (that is, whether it is a reasonable description in the real world) to filter out potentially unreasonable content.
>
> + However, we removed the **rationality judgment** step as we found that it does not significantly affect the training performance of the model. After using powerful LLMs to generate sufficient text data, the impact of noise in text data is negligible, which is also demonstrated by the related experiments in the PVP [1] (PVP [1] compares the performance of publicly annotated text data with those generated by LLMs, and the performance difference is small). Moreover, rationality judgment increases the cost of LLMs generation. We conduct the following experiment, further filtering out 300,000 high-quality text description data through rationality judgment:
>
> | Methods                   | K400 | MSCOCO | ESC50
> | ------- | ------- | ------- | ------- |
> | TaAM-CPT (300k)           | (55.2, 80.4) | 68.1 | 94.2
> | TaAM-CPT (cleaned 300k)   | (55.4, 80.1) | 68.0 | 93.9
>
> + We believe this may be because the large-scale pretraining data of pretrained models may also contain too much noise, hallucinations, etc., which desensitizes the model to noisy text.
>
> ### **Weakness 3: is there an information leak as inter-modal Learning based on validation performance?**
> + During the training of TaAM-CPT, relying on the validation performance is just a simple and direct strategy that we employ. Actually, besides the validation performance, other factors such as **training performance, training loss, convergence speed**, etc, can all serve as criteria for determining weak modalities. For example, the training loss for each modality is as follows:
>
> | Training epoch  | K400 | MSCOCO | ESC50 |
> | -------  | ------- | ------- | ------- |
> | Epoch-00 | 487.3 | 135.4 | 53.2 |
> | Epoch-03 | 386.1 | 105.3 | 14.2 |
> | Epoch-06 | 298.7 | 87.5 | 8.7 |
> | Epoch-09 | 287.5 | 83.5 | 6.9 |
> + We can see that the training loss for the video modality is the highest all the time, which may be due to the strong correlations among categories. Therefore, in the experimental benchmarks, the video modality is always considered as the weak modality.
> + Furthermore, as shown in Table 7 in the manuscript, if image or audio modalities are treated as weak modalities, it will decrease the performance of all modalities.
>
> ### **Weakness 4: Should TaAM-CPT be trained individually based on the validation performance of each dataset?**
> + Determining weak modalities is based on the overall performance across all datasets within each modality. For instance, the average performance of the video modality on the k400, k600, and k700 datasets. Hence, there is no need to train TaAM-CPT separately for each dataset. Moreover, we can also determine the weak modality by using the average training performance, and average training loss, rather than relying on the validation performance.
>
> [1]. TAI++:Text as Image for Multi-Label Image Classification by Co-Learning Transferable Prompt. IJCAI 2024

---

> ### Author Response · Authors · 2024-11-21
> **Official Response by Authors (2/2)**
>
> ### **Weakness 5: Experiments about improving the inference speed.**
> + We conduct an experimental analysis of inference cost with TAI-DPT [1] and PVP [2] on the MSCOCO2014 dataset, as shown in the table below:
>
> |  MSCOCO2014  | TAI-DPT | PVP | TaAM-CPT |
> | ------- | ------- | ------- | ------- |
> | Testing time | 8.3min | 8.9min | 2.8min |
>
> + TAI-DPT [1] designs multi-granularity prompts (global prompt and local prompt), while PVP [2] designs pseudo-visual prompt and text prompt, and both require prompt encoding processes. TaAM-CPT represents each category with a class-specific prompt, avoiding the prompt encoding processes,  and presenting significant advantages in terms of inference time.
>
> ### **Weakness 6: Compare the performance difference between TaAM-CPT and MLLM.**
> + The main difference between TaAM-CPT and MLLMs is that our work only uses LLMs to generate text training data, LLMs are not employed in the training process.
>
> + MLLMs are generative models, whose aim is to generate high-quality images, videos, or audio given a text prompt, rather than performing classification or regression. The performance advantage of TaAM-CPT stems from its class-specific prompt design and training with text data.
>
> ### **Weakness 7: Exhaustive permutation of many labels is time-consuming and may degrade model quality.**
> + In our experiments, we do not traverse all possible permutations of labels, instead, we randomly sample a portion of the text data for training.
>
> + For any category of any modality, generating 500~1000 text descriptions is sufficient. As shown in Figure 7 in the manuscript, additional training data does not further improve the performance.
>
> ### **Weakness 8: The burden of training intra- and inter-learning with more modalities and labels.**
> + We conduct theoretical analysis in Sec 4.4 in the manuscript and take the bert-base model as an example, presenting the feasibility of TaAM-CPT for extending unlimited modalities and categories. We believe TaAM-CPT is still a recommended choice when pursuing unlimited modalities or categories.
>
> + Adopt the Bert-base model with 110MB parameters as a reference. TaAM-CPT initializes each prompt with a 512-d vector, meaning one class prompt occupies 512 parameters. Therefore, for n modalities, TaAM-CPT can accommodate approximately 110,000,000512n class prompts. For example, for 10 modalities, the prompt pool size can reach 21,484, which is sufficient to cover the common categories. We can see that the training cost of training 10 modalities with 21,484 labels is comparable with the finetuning Bert-Base model. Moreover, TaAM-CPT directly updates the prompt without continual backpropagation paths.
>
> [1]. Texts as Images in Prompt Tuning for Multi-Label Image Recognition. CVPR 2023
>
> [2]. TAI++:Text as Image for Multi-Label Image Classification by Co-Learning Transferable Prompt. IJCAI 2024

---

> > ### Comment · Reviewer_25FQ · 2024-11-27
> >
> > Thank the authors for addressing some of my concerns, but I still think the methodology is a bit fragile, e.g., “intermodal instructional training”. Therefore, I am keeping my score.

---

> ### Comment · Area_Chair_Q33N · 2024-11-27
>
> Dear reviewer,
>
> Today is the last day for reviewers to ask questions to authors. Did the authors' rebuttal address your concern? Do you have any additional questions?

---

### Official Review · Reviewer_ngqX · 2024-11-03

**Soundness:** 2
**Presentation:** 2
**Contribution:** 2
**Rating:** 5
**Confidence:** 3

**Summary:**

This paper focuses on representation in a multimodal setting, introducing a method to build class-specific prompts across three types of inputs: image, video, and audio. The training process is efficient, relying only on a prompt pool as the training parameter and using pretrained models like CLIP, CLAP, and LLM to eliminate the need for complex data collection. However, the experiences discussed in this paper are limited to relatively simple classification problems involving images, videos, and audio. This hinders my ability to fully understand the potential of these methods.

**Strengths:**

1. This paper focuses on representation in a multimodal setting, which is a interesting and important fields. This paper also utilize important tools of contrastive loss for inter-model learning and ranking loss for intra-model learning.
2. The training process is efficient, relying only on a prompt pool as the training parameter and using pretrained models like CLIP, CLAP, and LLM to eliminate the need for complex data collection.
3. The experiences include both objective metrics and visual figures to help understand the effectiveness of the new methods.

**Weaknesses:**

The main issue lies in the novelty and practical functionality for this fields:
1. The contrastive loss and ranking loss are not new; they have been applied to multimodal representation learning [1, 2] for some time.
2. The application of this paper is currently limited to relatively simple classification tasks, for which many existing tools perform well. I encourage the authors to include additional tasks, such as conditional image or audio generation using the novaly class.
3. The prompt pool is limited at the beginning, so what if the user wants to add a new concept? Although the pipeline can introduce new concepts, the issue is how quickly this process can be compared to the overall training of the method. I encourage the authors to provide more empirical results or theoretical analysis on the time complexity of adding new concepts in comparison to initial training time.

[1] Wang Z, Zhao Y, Huang H, et al. Connecting multi-modal contrastive representations[J]. Advances in Neural Information Processing Systems, 2023, 36: 22099-22114.

[2] Zheng L, Jing B, Li Z, et al. Heterogeneous contrastive learning for foundation models and beyond[C]//Proceedings of the 30th ACM SIGKDD Conference on Knowledge Discovery and Data Mining. 2024: 6666-6676.

**Questions:**

See the weaknesses.

---

> ### Author Response · Authors · 2024-11-21
> **Official Response by Authors (1/1)**
>
> Thanks for your valuable suggestions and we will try to address your concerns as follows. We are eager to engage in a more detailed discussion with you.
> ### **Weakness 1: The contrastive and ranking losses are not new.**
> + We understand your concerns about the novelty. The innovation of TaAM-CPT does not lie in the choice of loss function, both contrastive and ranking losses are merely optimization objectives for TaAM-CPT. The key novelty of TaAM-CPT is its minimalist training process, which relies solely on text data without any modality-specific annotated data. This enhances the applicability of multimodal pre-training models in scenarios with scarce data.
>
> + TaAM-CPT simplifies prompt design by using randomly initialized vectors to represent all categories, thus avoiding complex prompt design and encoding processes, and significantly reducing training costs. Moreover, as a versatile representation model, TaAM-CPT can be extended to accommodate unlimited modalities and categories. This flexibility and scalability are not present in existing methods.
>
> ### **Weakness 2: Include additional tasks, such as conditional image or audio generation using the novel class.**
> + We agree with your suggestions regarding the practical application. TaAM-CPT primarily focuses on the classification task in zero-shot scenarios, using only textual data instead of annotated modality-specific data. It achieves significant performance across various modalities and datasets through simplified class-specific prompts and multimodal pre-training models.
>
> + However, we also fully recognize the limitations of classification tasks in real-world applications. In scenarios with intricate generation tasks, the model needs to be able to generate modalities such as video, images, or audio. Therefore, we will continue exploring generation abilities in our future research to enable TaAM-CPT to also generate modalities corresponding to novel categories, thereby expanding the scope and utility of TaAM-CPT in various applications.
>
> ### **Weakness 3: Provide empirical results on the time complexity of adding new concepts in comparison to initial training time.**
> + The trainable parameters of TaAM-CPT are the class-specific prompts. Therefore, for simplicity, we random samples **K400 with 30 video labels** and **MSCOCO with 30 image labels** as the initial label set. The novel concepts consist of additional labels **(K400 with 20 video labels, MSCOCO with 20 image labels)**, and the novel audio modality **(ESC50 with 20 labels)**, with each label containing **500** text descriptions using LLMs. The comparison of training time results is shown below:
>
> | Modality_Labels   |  |  |  |
> | ------- | ------- | ------- | ------- |
> | K400_Normal_30    | Initial training **(58.1, 82.3)** | Continue training **(58.4, 82.5)** | Frozen **(58.1, 82.3)** |
> | MSCOCO_Normal_30  | Initial training **(65.4)** | Continue training **(65.6)** | Frozen **(65.4)** |
> | K400_Novel_20     | Initial training **(56.3)** | Novel training **(56.7)** | Novel training **(56.5)** |
> | MSCOCO_Novel_20   | Initial training **(69.2)** | Novel training **(68.8)** | Novel training **(69.3)** |
> | ESC50_Novel_20    | Initial training **(92.1)** | Novel training **(92.5)** | Novel training **(93.0)** |
> | **Training time** | **28min** | **30min** | **17min** |
>
> + Among them, **(K400_Novel_20, MSCOCO_Novel_20, ESC50_Novel_20)** represent the added new labels and new modality. We can see that for the normal labels **(K400_Normal_30, MSCOCO_Normal_30)**, whether they continue training **(Continue training)** or are frozen **(Frozen)**, neither affects the learning of the new modality labels and maintain the performance that matches the initial joint training **(Initial training)**. For the novel modalities and labels, their **Novel training** performances are still comparable with **Initial training**.

---

> ### Author Response · Authors · 2024-11-23
> **Looking forward to your feedback.**
>
> Dear Reviewer, we are looking forward to your professional suggestions and hope to receive your guidance to further discuss and refine the contents of the work. We are eager for your response. Thank you.

---

> ### Comment · Area_Chair_Q33N · 2024-11-27
>
> Dear reviewer,
>
> Today is the last day for reviewers to ask questions to authors. Did the authors' rebuttal address your concern? Do you have any additional questions?

---

### Meta-Review · Area_Chair_Q33N · 2024-12-23

**Metareview:**

This paper was reviewed by four experts in the field, and reviewers unanimously agreed to reject the paper. The authors' rebuttal didn't flip reviewers' initial ratings. The AC read the paper, reviews, and rebuttal carefully but didn't find strong reasons to overturn the reviewers' consensus.

**Additional Comments On Reviewer Discussion:**

The authors' rebuttal didn't flip reviewers' initial ratings. The AC read the paper, reviews, and rebuttal carefully but didn't find strong reasons to overturn the reviewers' consensus.

---

### Decision · Program_Chairs · 2025-01-22

Reject